# Convergence Bound and Critical Batch Size of Muon Optimizer

## Abstract

Muon, a recently proposed optimizer that leverages the inherent matrix structure of neural network parameters, has demonstrated strong empirical performance, indicating its potential as a successor to standard optimizers such as AdamW. This paper presents theoretical analysis to support its practical success. We provide convergence proofs for Muon across four practical settings, systematically examining its behavior with and without the inclusion of Nesterov momentum and weight decay. Our analysis covers the standard configuration using both, thereby elucidating its real-world performance. We then demonstrate that the addition of weight decay yields strictly tighter theoretical bounds and clarify the interplay between the weight decay coefficient and the learning rate. Finally, we derive the critical batch size for Muon that minimizes the computational cost of training. Our analysis identifies the hyperparameters governing this value, and our experiments validate the corresponding theoretical findings across workloads including image classification and language modeling task.

## 1 Introduction

Optimization algorithms are fundamental to the training of deep neural networks (DNNs). Since the introduction of stochastic gradient descent (SGD) (Robbins & Monro, 1951), numerous optimizers have been proposed to accelerate convergence. Among these, adaptive gradient algorithms such as Adam (Kingma & Ba, 2015) and its subsequent variant AdamW (Loshchilov & Hutter, 2019) have emerged as the de facto standard in modern deep learning, valued for their rapid convergence and robust performance across a wide range of tasks. A common characteristic of these widely used first-order optimizers is that they treat the weight parameters of neural networks, which are inherently matrices or higher-order tensors, as high-dimensional vectors. However, this vector-based perspective, while effective, disregards the underlying geometric and algebraic structure within the parameter matrices.

The recently proposed Muon optimizer (Jordan et al., 2024) introduces a distinct paradigm that departs from the conventional vector-based viewpoint. The core idea of Muon is to preserve and leverage the intrinsic matrix structure of the network parameters. Instead of using the gradient vector, Muon computes its search direction by orthogonalizing the gradient momentum matrix. Specifically, for a given momentum matrix $C_t \in \mathbb{R}^{m \times n}$, the search direction $O_t \in \mathbb{R}^{m \times n}$ is the orthogonal matrix that is closest in Frobenius norm, found by solving

$$O_t := \underset{O \in \{O \in \mathbb{R}^{m \times n} : O^\top O = I_n\}}{\mathrm{argmin}} \|O - C_t\|_{\mathrm{F}}, \tag{1}$$

where $I_n$ denotes a $n \times n$ identity matrix. As established by classic matrix theory, if the singular value decomposition (SVD) of $C_t$ is $U_t S_t V_t^\top$, the optimal search direction is simply $O_t = U_t V_t^\top$, where $U_t \in \mathbb{R}^{m \times r}, S_t \in \mathbb{R}^{r \times r}, V_t \in \mathbb{R}^{n \times r}$, and $r > 0$ is the rank of $C_t$. This process of gradient orthogonalization is aimed at finding a search direction that is independent of the gradient's magnitude, potentially leading to more stable and effective training dynamics. While SVD provides an exact analytical solution, its high computational cost renders it impractical for the large matrices found in modern DNNs. As detailed by

Bernstein (2025), the key computational innovation of Muon is the use of the Newton-Schulz iteration (Bernstein & Newhouse, 2024; Higham, 2008; Björck & Bowie, 1971; Kovarik, 1970), a classic and remarkably efficient numerical method, to approximate this orthogonalization. This iterative algorithm enables Muon to compute the search direction without performing an explicit SVD, making it a computationally feasible optimizer for large-scale applications. This elegant fusion of a novel optimization perspective with a powerful numerical technique positions Muon as a promising, theoretically grounded alternative to existing optimizers.

Several studies have reported the strong empirical performance of Muon. Jordan et al. (2024) showed that Muon outperforms Shampoo (Gupta et al., 2018) and SOAP (Vyas et al., 2025), both on a per-step and wall-clock basis. Liu et al. (2025b) showed that by dividing the parameters and updating them with SOAP and Muon, optimization performance equivalent to or better than SOAP can be achieved while substantially reducing memory usage. Liu et al. (2025a) demonstrated that Muon is effective for training large-scale LLMs and suggested it has the potential to replace AdamW as the standard optimizer. AI et al. (2025) showed that Muon expands AdamW's Pareto frontier on the compute-time plane, enlarging the practitioner's flexibility in resource allocation. However, the theoretical understanding of Muon's convergence behavior remains underdeveloped, and a formal justification for its strong performance over AdamW is still lacking. This work aims to bridge this gap by providing a rigorous convergence analysis of Muon.

Furthermore, since the batch size is a critical hyperparameter for managing computational costs in large-scale training, we also consider Muon's critical batch size. This is defined as the batch size that minimizes the computational cost of training. In other words, the critical batch size is the point at which further increases in batch size yield diminishing returns in hardware throughput (i.e., the number of samples processed per unit time). By understanding and utilizing the critical batch size, one can maximize GPU utilization, thereby shortening training times and reducing overall computing costs. Therefore, to theoretically understand Muon and maximize its performance, analyzing its critical batch size is essential. Following previous studies, we aim to clarify the critical batch size of Muon.

Table 1: Upper bound of $\frac{1}{T}\sum_{t\in[T]}\mathbb{E}\left[\|\nabla f(W_t)\|_{\mathrm{F}}\right]$. See Section 2 for notation.

|  | w/o weight decay | w/ weight decay |
|---|---|---|
| w/o Nesterov | $\mathcal{O}\left(\frac{1}{T} + \frac{1-\beta}{b} + \sqrt{\frac{1-\beta}{b}} + n\right)$ (Theorem 3.1(i)) | $\mathcal{O}\left(\frac{1}{T} + \left(1 - \beta + \frac{\lambda}{2}\right)\frac{1}{b} + \sqrt{\frac{1-\beta}{b}} + n\right)$ (Theorem 3.2(i)) |
| w/ Nesterov | $\mathcal{O}\left(\frac{1}{T} + \left(\beta + \frac{1}{2}\right)(1-\beta)\frac{1}{b} + \beta\sqrt{\frac{1-\beta}{b}} + n\right)$ (Theorem 3.1(ii)) | $\mathcal{O}\left(\frac{1}{T} + \left\{\left(\beta + \frac{1}{2}\right)(1-\beta) + \frac{\lambda}{2}\right\}\frac{1}{b} + \beta\sqrt{\frac{1-\beta}{b}} + n\right)$ (Theorem 3.2(ii)) |

Our main contributions are as follows:

- We present a convergence analysis for four variants of Muon—with and without Nesterov momentum and with and without weight decay (Theorems 3.1 and 3.2). For each variant, we establish an upper bound on the average expected gradient norm, summarized in Table 1. The variant combining both Nesterov momentum and weight decay is of particular interest as it mirrors common practical settings. Our analysis thus offers direct insights into Muon's real-world behavior.

- We prove that incorporating weight decay yields tighter theoretical bounds on both the parameter and gradient norms (Propositions 3.1 and 3.2) and present experimental results to support this finding. We also show that for Muon to converge, the learning rate $\eta$ and weight decay coefficient $\lambda$ must satisfy $\eta \leq \frac{1}{\lambda}$, a condition supported by our experimental results (see Figure 1).

- We derive the critical batch size for the four Muon variants (Proposition 4.3). For example, we show that the critical batch size of Muon with Nesterov momentum and weight decay is given by

$$b_{\mathrm{Muon}}^{\star} = \frac{\left\{(2\beta + 1)(1 - \beta) + \lambda\right\}\sigma^2}{\epsilon},$$

where $\beta \in (0,1]$ is momentum, $\lambda > 0$ is the weight decay coefficient, $\sigma^2 > 0$ is the variance of the stochastic gradient, and $\epsilon > 0$ is the threshold or stopping condition. While not fully predictive due to its reliance on several unknown parameters, our theoretical analysis produces a formula for the critical batch size that successfully identifies the hyperparameters governing this value, as validated by our empirical results (see Figure 4).

## 2  Preliminaries

### 2.1  Notations and Definition

Let $\mathbb{N}$ be the set of nonnegative integers. For $p \in \mathbb{N} \backslash \{0\}$, define $[p] := \{1, 2, \ldots, p\}$. Let $\mathbb{R}^d$ be a $d$-dimensional Euclidean space. We use lowercase letters for scalars (e.g., $a \in \mathbb{R}$), bold lowercase letters for vectors (e.g., $\boldsymbol{a} \in \mathbb{R}^d$), and uppercase letters for matrices (e.g., $A \in \mathbb{R}^{m \times n}$). $\boldsymbol{a}^\top \in \mathbb{R}^{1 \times d}$ and $A^\top \in \mathbb{R}^{n \times m}$ denote the transposes of $\boldsymbol{a} \in \mathbb{R}^d$ and $A \in \mathbb{R}^{m \times n}$, respectively. For a square matrix $A = (a_{ij}) \in \mathbb{R}^{n \times n}$, the trace is defined as $\mathrm{tr}(A) := \sum_{i=1}^n a_{ii}$. For all vectors $\boldsymbol{x}, \boldsymbol{y} \in \mathbb{R}^d$, the Euclidean inner product is defined as $\langle \boldsymbol{x}, \boldsymbol{y} \rangle_2 := \boldsymbol{x}^\top \boldsymbol{y}$ and the Euclidean norm is defined as $\|\boldsymbol{x}\|_2 := \sqrt{\langle \boldsymbol{x}, \boldsymbol{x} \rangle_2}$. For all matrices $A, B \in \mathbb{R}^{m \times n}$, the Frobenius inner product is defined as $\langle A, B \rangle_{\mathrm{F}} := \mathrm{tr}(A^\top B)$, and the Frobenius norm defined as $\|A\|_{\mathrm{F}} := \sqrt{\langle A, A \rangle_{\mathrm{F}}}$. The model is parameterized by a matrix $W \in \mathbb{R}^{m \times n}$ ($m \geq n$), which is optimized by minimizing the empirical loss function $f(W) := \frac{1}{N} \sum_{i \in [N]} f_i(W)$, where $N \in \mathbb{R}$ is the number of training samples and $f_i(W)$ denotes the loss associated with the $i$-th training sample $\boldsymbol{z}_i$ ($i \in [N]$). We define $W^\star := \mathrm{argmin}_{W \in \mathbb{R}^{m \times n}} f(W)$. Let $\xi$ be a random variable that does not depend on $W \in \mathbb{R}^{m \times n}$, and let $\mathbb{E}_\xi[X]$ denote the expectation with respect to $\xi$ of a random variable $X$. $\xi_{t,i}$ is a random variable generated from the $i$-th sampling at time $t$, and $\xi_{t,i}$ and $\xi_{t,j}$ are independent ($i \neq j$). $\boldsymbol{\xi}_t := (\xi_{t,1}, \xi_{t,2}, \ldots, \xi_{t,b})^\top$ is independent of sequence $(W_k)_{k=0}^t \subset \mathbb{R}^{m \times n}$ generated by Muon (Algorithm 1), where $b$ ($\leq N$) is the batch size. The independence of $\boldsymbol{\xi}_0, \boldsymbol{\xi}_1, \ldots$ allows us to define the total expectation $\mathbb{E}$ as $\mathbb{E} = \mathbb{E}_{\boldsymbol{\xi}_0} \mathbb{E}_{\boldsymbol{\xi}_1} \cdots \mathbb{E}_{\boldsymbol{\xi}_t}$. Let $\mathsf{G}_\xi(W)$ be the stochastic gradient of $f(\cdot)$ at $W \in \mathbb{R}^{m \times n}$. The mini-batch $\mathcal{S}_t$ consists of $b$ samples at time $t$, and the mini-batch stochastic gradient of $f(W_t)$ for $\mathcal{S}_t$ is defined as $\nabla f_{\mathcal{S}_t}(W_t) := \frac{1}{b} \sum_{i \in [b]} \mathsf{G}_{\xi_{t,i}}(W_t) = \frac{1}{b} \sum_{i \in \mathcal{S}_t} \nabla f_i(W_t)$.

Algorithm 1 presents the most common variant of Muon, which incorporates Nesterov momentum and weight decay. Our implementation of Nesterov momentum and decoupled weight decay (Loshchilov & Hutter, 2019) follows the original formulations of (Jordan et al., 2024). Muon optimizers with both Nesterov momentum and weight decay are often used in practice (e.g., Jordan et al., 2024; AI et al., 2025). NewtonSchulz5($\cdot$) receives $C_t$, sets $X_0 := C_t / \|C_t\|_{\mathrm{F}}$, performs the following iterations from $k = 0$ to 4, and returns $X_5$.

$$X_{k+1} := aX_k + b(X_k X_k^\top)X_k + c(X_k X_k^\top)^2 X_k,$$

where $a = 3.4445, b = -4.7750, c = 2.0315$. The sequence $(X_k)_{k \in \mathbb{N}}$ converges to $O_t$ defined as Eq. (1) (Bernstein & Newhouse, 2024; Higham, 2008). Our theoretical analysis assumes that $O_t := $ NewtonSchulz5($C_t$) satisfies Eq. (1). Therefore,

---

**Algorithm 1** Muon

**Require:** $\eta, \lambda > 0, \beta \in [0,1), M_{-1} := \mathbf{0}, W_0 \in \mathbb{R}^{m \times n}$
  **for** $t = 0$ to $T - 1$ **do**
    $M_t := \beta M_{t-1} + (1 - \beta)\nabla f_{\mathcal{S}_t}(W_t)$
    **if** (Nesterov = True) **then**
      $C_t := \beta M_t + (1 - \beta)\nabla f_{\mathcal{S}_t}(W_t)$
    **else**
      $C_t := M_t$
    **end if**
    $O_t := $ NewtonSchulz5($C_t$)
    **if** (weight decay = True) **then**
      $W_{t+1} := (1 - \eta\lambda)W_t - \eta O_t$
    **else**
      $W_{t+1} := W_t - \eta O_t$
    **end if**
  **end for**
  **return** $W_T$

---

there may be a gap between the Muon we consider theoretically and its practical implementation. On the other hand, according to Shen et al. (2025), the experimental performance of the SVD-based Muon is equivalent to that of the Newton-Schulz-based version, with the main difference being the higher computational cost of the SVD procedure. That is, experimentally, no gap has been demonstrated.

### 2.2  Assumptions

We make the following standard assumptions:

**Assumption 2.1.** *The function $f \colon \mathbb{R}^{m \times n} \to \mathbb{R}$ is $L$-smooth, i.e., for all $A, B \in \mathbb{R}^{m \times n}$,*

$$\|\nabla f(A) - \nabla f(B)\|_{\mathrm{F}} \le L\|A - B\|_{\mathrm{F}}.$$

**Assumption 2.2.** (i) *For all $t$ and all $i$,*

$$\mathbb{E}_{\xi_{t,i}} \left[ \mathsf{G}_{\xi_{t,i}}(W_t) \right] = \nabla f(W_t).$$

(ii) *There exists a nonnegative constant $\sigma^2$ such that, for all $t$ and all $i$,*

$$\mathbb{E}_{\xi_{t,i}} \left[ \|\mathsf{G}_{\xi_{t,i}}(W_t) - \nabla f(W_t)\|_{\mathrm{F}}^2 \right] \le \sigma^2.$$

## 3 Analysis of Muon's convergence

### 3.1 Muon without weight decay

We now present a convergence analysis of Muon (Algorithm 1) without weight decay. The proofs of Theorem 3.1(i) and (ii) are in Appendix B.

**Theorem 3.1.** *Suppose Assumptions 2.1 and 2.2 hold. Then, for all $t \in \mathbb{N}$,*

(i) for Muon without Nesterov and without Weight Decay,

$$\frac{1}{T} \sum_{t=0}^{T-1} \mathbb{E}\left[ \|\nabla f(W_t)\|_{\mathrm{F}} \right] \le \frac{f(W_0)}{\eta T} + \frac{\Delta^2 + 2\sqrt{2r}\Delta}{(1-\beta)T} + \frac{(1-\beta)\sigma^2}{b} + \nu\sqrt{\frac{r\sigma^2}{b}} + 2\gamma(\sqrt{r} + \gamma) + \frac{1 + L\eta}{2}n$$

$$= \mathcal{O}\left( \frac{1}{T} + \frac{1-\beta}{b} + \sqrt{\frac{1-\beta}{b}} + n \right),$$

(ii) for Muon with Nesterov and without Weight Decay,

$$\frac{1}{T} \sum_{t=0}^{T-1} \mathbb{E}\left[ \|\nabla f(W_t)\|_{\mathrm{F}} \right] \le \frac{f(W_0)}{\eta T} + \frac{\beta(\Delta^2 + 2\sqrt{2r}\Delta)}{(1-\beta)T} + \frac{\bar{\beta}\sigma^2}{b} + (\beta\nu + 1 - \beta)\sqrt{\frac{r\sigma^2}{b}} + 2\gamma\beta(\sqrt{r} + \gamma) + \frac{1 + L\eta}{2}n$$

$$= \mathcal{O}\left( \frac{1}{T} + \left\{ \left( \beta + \frac{1}{2} \right)(1-\beta) \right\} \frac{1}{b} + \sqrt{\frac{1-\beta}{b}} + n \right),$$

*where $\Delta := \|M_0 - \nabla f(W_0)\|_{\mathrm{F}}^2$, $\bar{\beta} := \frac{(2\beta+1)(1-\beta)}{2}$, $\nu := \sqrt{2(1-\beta)}$, $\gamma := \frac{L\eta\sqrt{n}}{1-\beta}$, and $r := \max\limits_{0 \le t \le T-1} \mathrm{rank}(C_t - \nabla f(W_t))$.*

Theorem 3.1 show that Muon, both with and without Nesterov momentum, achieves similar upper bounds on convergence. However, some terms in the upper bound are slightly smaller when Nesterov momentum is used. Our bounds contain terms independent of $T$, particularly those involving $n$, so we cannot guarantee convergence to stationary point. This is due to $\|O_t\|_{\mathrm{F}}^2 \le n$, and cannot be avoided using standard proof techniques. Since the bounds depend on the parameter dimension $n$, we cannot explain why Muon performs well even in large-scale numerical experiments. Resolving this issue is undoubtedly important future work.

### 3.2 Muon with weight decay

The following proposition establishes a key result for Muon (Algorithm 1) with weight decay. The proofs of Propositions 3.1 and 3.2 are in Appendix C.

**Proposition 3.1.** *Suppose Assumptions 2.1 and 2.2 hold, and that Muon is run with $\eta \le \frac{1}{\lambda}$. Then, for all $t \in \mathbb{N}$,*

$$\|W_t\|_{\mathrm{F}} \le \begin{cases} (1 - \eta\lambda)^t \|W_0\|_{\mathrm{F}} + \frac{\sqrt{n}}{\lambda} & \text{if } \eta < \frac{1}{\lambda}, \\ \frac{\sqrt{n}}{\lambda} & \text{if } \eta = \frac{1}{\lambda}. \end{cases}$$

**Proposition 3.2.** *Suppose Assumptions 2.1 and 2.2 hold, and that Muon is run with $\eta \leq \frac{1}{\lambda}$. Then, for all $t \in \mathbb{N}$,*

$$\|\nabla f(W_t)\|_{\mathrm{F}} \leq \begin{cases} L(1 - \eta\lambda)^t \|W_0\|_{\mathrm{F}} + \frac{L}{\lambda} + L\|W^\star\|_{\mathrm{F}} & \text{if } \eta < \frac{1}{\lambda}, \\ \frac{L}{\lambda} + L\|W^\star\|_{\mathrm{F}} & \text{if } \eta = \frac{1}{\lambda}. \end{cases}$$

Proposition 3.1 establishes that when $\eta \leq \frac{1}{\lambda}$, weight decay ensures the parameter norm remains almost surely bounded. Furthermore, the upper bound decreases monotonically with $t$, converging to $\frac{\sqrt{n}}{\lambda}$ as $t \to \infty$. The bound is minimized uniformly across all $t$ when $\eta = \frac{1}{\lambda}$. Proposition 3.2 extends the result of Proposition 3.1 to the full gradient norm, which is likewise almost surely bounded. This bound decreases monotonically with $t$, converging to $\frac{L}{\lambda} + L\|W^\star\|_{\mathrm{F}}$ as $t \to \infty$. From these results, Corollary 3.1 establishes an almost surely bound of Muon. In both cases, the upper bounds are minimized when $\eta = \frac{1}{\lambda}$.

**Corollary 3.1.** *Suppose Assumptions 2.1 and 2.2 hold and $\eta \leq \frac{1}{\lambda}$. Then, for all $T \in \mathbb{N}$,*

$$\frac{1}{T} \sum_{t=0}^{T-1} \|\nabla f(W_t)\|_{\mathrm{F}} \leq \begin{cases} \frac{L\|W_0\|_{\mathrm{F}}}{\eta\lambda T} + \frac{L}{\lambda} + L\|W^\star\|_{\mathrm{F}} & \text{if } \eta < \frac{1}{\lambda}, \\ \frac{L}{\lambda} + L\|W^\star\|_{\mathrm{F}} & \text{if } \eta = \frac{1}{\lambda}. \end{cases}$$

While these results suggest that a larger weight decay $\lambda$ yields a tighter bound, the condition $\eta \leq \frac{1}{\lambda}$ necessitates a smaller learning rate $\eta$. These desirable properties stem from the fact that Muon's search direction is inherently bounded. A key advantage of this feature is that our analysis does not rely on the common-and often restrictive-assumption of bounded gradients.

The following is a convergence analysis of Muon (Algorithm 1) with weight decay (the proofs of Theorems 3.2(i) and (ii) are in Appendix C).

**Theorem 3.2.** *Suppose Assumptions 2.1 and 2.2 hold, and that Muon is run with $\eta \leq \frac{1}{\lambda}$. Then, for all $t \in \mathbb{N}$,*

(i) for Muon without Nesterov and with Weight Decay,

$$\frac{1}{T} \sum_{t=0}^{T-1} \mathbb{E}\left[\|\nabla f(W_t)\|_{\mathrm{F}}\right] \leq \frac{f(W_0) + \rho\|W_0\|_{\mathrm{F}}^2}{\eta T} + \frac{\Delta^2 + 2\sqrt{2r}\Delta}{(1-\beta)T} + \left(1 - \beta + \frac{\lambda}{2}\right)\frac{\sigma^2}{b}$$

$$+ \nu\sqrt{\frac{r\sigma^2}{b}} + 2\gamma(\sqrt{r} + \gamma) + (1 + L\eta)n + \frac{\rho n}{\lambda} + \frac{\lambda D_0^2}{2}$$

$$= \mathcal{O}\left(\frac{1}{T} + \left(1 - \beta + \frac{\lambda}{2}\right)\frac{1}{b} + \sqrt{\frac{1-\beta}{b}} + n\right),$$

(ii) for Muon with Nesterov and with Weight Decay,

$$\frac{1}{T} \sum_{t=0}^{T-1} \mathbb{E}\left[\|\nabla f(W_t)\|_{\mathrm{F}}\right] \leq \frac{f(W_0) + \rho\|W_0\|_{\mathrm{F}}^2}{\eta T} + \frac{\beta(\Delta^2 + 2\sqrt{2r}\Delta)}{(1-\beta)T} + \left(\bar{\beta} + \frac{\lambda}{2}\right)\frac{\sigma^2}{b}$$

$$+ (\beta\nu + 1 - \beta)\sqrt{\frac{r\sigma^2}{b}} + 2\gamma\beta(\sqrt{r} + \gamma) + (1 + L\eta)n + \frac{\rho n}{\lambda} + \frac{\lambda D_0^2}{2}$$

$$= \mathcal{O}\left(\frac{1}{T} + \left\{\left(\beta + \frac{1}{2}\right)(1 - \beta) + \frac{\lambda}{2}\right\}\frac{1}{b} + \sqrt{\frac{1-\beta}{b}} + n\right),$$

*where $\Delta := \|M_0 - \nabla f(W_0)\|_{\mathrm{F}}^2$, $\bar{\beta} := \frac{(2\beta+1)(1-\beta)}{2}$, $\nu := \sqrt{2(1-\beta)}$, $\gamma := \frac{L\eta\sqrt{n}}{1-\beta}$, $\rho := \frac{1+2(1+L\eta)\lambda}{2}$, $D_0 := L\left(\|W_0\|_{\mathrm{F}} + \frac{\sqrt{n}}{\lambda} + \|W^\star\|_{\mathrm{F}}\right)$, and $r := \max_{0 \leq t \leq T-1} \mathrm{rank}(C_t - \nabla f(W_t))$.*

Similar conclusions follow from Theorems 3.2, which again demonstrate a modest advantage from incorporating Nesterov momentum. These results build on Propositions 3.1 and 3.2 and therefore inherit the assumption that $\eta \leq \frac{1}{\lambda}$. In other words, for Muon with weight decay to attain the stated convergence rate, it must satisfy $\eta \leq \frac{1}{\lambda}$. Practically speaking, since the weight decay coefficient $\lambda$ is typically less than 1, this assumption is realistic and does not materially constrain the choice of learning rate.

# 4 Analysis of Muon's critical batch size

Our theoretical analysis characterizes the critical batch size as a function of the gradient variance $\sigma^2$ and optimization hyperparameters. While explicitly modeling the dependence of $\sigma^2$ on model width or depth is beyond the scope of this single-matrix analysis, our results establish the fundamental relationship $b^\star \propto \sigma^2$. This suggests that scaling behaviors observed in larger models are mediated through changes in their gradient noise properties. Indeed, recent large-scale empirical studies report that Muon remains efficient at increasingly large batch sizes for large language models (Liu et al., 2025a); our theory supports this observation, predicting that if larger models entail distinct gradient variance characteristics, the critical batch size will shift accordingly. With this motivation, we now formalize the notion of the critical batch size used in our analysis.

We next introduce the concept of the critical batch size, defined as the batch size that minimizes computational complexity. This complexity is measured in terms of the stochastic first-order oracle (SFO) complexity, which is the total number of stochastic gradient computations. Since the optimizer computes $b$ stochastic gradients per step, an optimizer that runs for $T$ steps with batch size $b$ incurs a total of $Tb$ SFO complexity. Empirically, for batch sizes up to a certain threshold $b^\star$ (the critical batch size), the number of training steps $T$ required to train a DNN scales inversely with $b$ (Shallue et al., 2019; Ma et al., 2018; McCandlish et al., 2018). Beyond $b^\star$, increasing the batch size yields diminishing returns in reducing $T$. The critical batch size is therefore the batch size that minimizes SFO complexity $Tb$. Prior work has shown that $b^\star$ depends on both the optimizer (Zhang et al., 2019) and dataset size (Zhang et al., 2025) and has established a theoretical framework for proving its existence and estimating its lower bound (Sato & Iiduka, 2023; Imaizumi & Iiduka, 2024). To analyze the critical batch size of Muon, we adopted this framework.

## 4.1 Relationship between batch size and number of steps needed for training

Suppose Assumptions 2.1 and 2.2 hold. Then, by Theorems 3.1 and 3.2, the following inequality holds:

$$\frac{1}{T} \sum_{t=0}^{T-1} \mathbb{E}\left[\|\nabla f(W_t)\|_{\mathrm{F}}\right] \leq \frac{X}{T} + \frac{Y}{b} + Z,$$

where $X, Y, Z > 0$ are nonnegative constants. Since $Y$ and $Z$ are constants independent of $T$, they do not decrease as the number of steps $T$ increases. Therefore, the upper bound of the gradient norm converges to $\frac{Y}{b} + Z$ as $T \to \infty$. To clarify the relationship between the batch size and the number of steps required for training, we exclude the term $Z$. Let $\epsilon > 0$ be an arbitrarily fixed threshold. When training is sufficiently complete, we assume that

$$\exists T, \exists b : \frac{X}{T} + \frac{Y}{b} < \epsilon, \tag{2}$$

where $\epsilon$ is not the threshold for the mean gradient norm. For details, see Remark 4.1. The relationship between $b$ and the number of steps $T_b$ satisfying Eq. (2) is as follows:

**Proposition 4.1.** *Suppose Assumptions 2.1 and 2.2 hold and let Muon be the optimizer under consideration. Then, $T_b$ defined by*

$$T_b := \frac{Xb}{\epsilon b - Y} < T \quad for \ \ b > \frac{Y}{\epsilon}, \tag{3}$$

*satisfies Eq. (2). In addition, the function $T_b$ defined by Eq. (3) is monotone decreasing and convex for $b > \frac{Y}{\epsilon}$.*

*Proof.* According to Eq. (3), Muon satisfies Eq. (2). For $b > \frac{Y}{\epsilon}$, we have

$$\frac{\mathrm{d}T_b}{\mathrm{d}b} = \frac{-XY}{(\epsilon b - Y)^2} \leq 0, \ \frac{\mathrm{d}^2 T_b}{\mathrm{d}b^2} = \frac{2XY\epsilon}{(\epsilon b - Y)^3} \geq 0.$$

Therefore, $T_b$ is monotone decreasing and convex for $b > \frac{Y}{\epsilon}$. This completes the proof. $\qquad\square$

## 4.2 Existence of a critical batch size

The critical batch size minimizes the computational complexity for training. Here, we use SFO complexity as a measure of computational complexity. Since the stochastic gradient is computed $b$ times per step, SFO complexity is defined as

$$T_b b = \frac{Xb^2}{\epsilon b - Y}. \tag{4}$$

The following theorem guarantees the existence of critical batch sizes that are global minimizers of $T_b b$ defined by Eq. (4).

**Proposition 4.2.** *Suppose that Assumptions 2.1 and 2.2 hold and consider Muon. Then, there exists*

$$b_{Muon}^\star := \frac{2Y}{\epsilon} \tag{5}$$

*such that $b_{Muon}^\star$ minimizes the convex function $T_b b$.*

*Proof.* From Eq. (5), we have that, for $b > \frac{Y}{\epsilon}$,

$$\frac{\mathrm{d}T_b b}{\mathrm{d}b} = \frac{Xb(\epsilon b - 2Y)}{(\epsilon b - Y)^2}, \ \frac{\mathrm{d}^2 T_b b}{\mathrm{d}b^2} = \frac{2XY^2}{(\epsilon b - Y)^3} \geq 0.$$

Hence, $T_b b$ is convex for $b > \frac{Y}{\epsilon}$, and

$$\frac{\mathrm{d}T_b b}{\mathrm{d}b} \begin{cases} < 0 & \text{if } b < b_{\mathrm{Muon}}^\star, \\ = 0 & \text{if } b = b_{\mathrm{Muon}}^\star = \frac{2Y}{\epsilon}, \\ > 0 & \text{if } b > b_{\mathrm{Muon}}^\star. \end{cases}$$

This completes the proof. $\qquad\square$

On the basis of Theorems 3.1 and 3.2 and Proposition 4.2, we derive the following proposition, which gives $b_{\mathrm{Muon}}^\star$.

**Proposition 4.3.** *Suppose Assumptions 2.1 and 2.2 hold. Then, for a given precision $\epsilon$, the critical batch size for Muon is as shown in Table 2.*

Table 2: Approximate critical batch size $b_{\mathrm{Muon}}^\star$ computed with $\beta = 0.95$ and $\lambda = 0.1$.

|  | w/o weight decay | w/ weight decay |
|---|---|---|
| w/o Nesterov | $\dfrac{2(1-\beta)\sigma^2}{\epsilon} \approx 0.1 \times \dfrac{\sigma^2}{\epsilon}$ | $\dfrac{\{2(1-\beta)+\lambda\}\sigma^2}{\epsilon} \approx 0.2 \times \dfrac{\sigma^2}{\epsilon}$ |
| w/ Nesterov | $\dfrac{(2\beta+1)(1-\beta)\sigma^2}{\epsilon} \approx 0.145 \times \dfrac{\sigma^2}{\epsilon}$ | $\dfrac{\{(2\beta+1)(1-\beta)+\lambda\}\sigma^2}{\epsilon} \approx 0.245 \times \dfrac{\sigma^2}{\epsilon}$ |

The results in Table 2 indicate that, for a given precision $\epsilon$, Muon's critical batch size is slightly larger when Nesterov momentum and weight decay are used. Furthermore, in all cases, $2Y$ becomes smaller as $\beta \to 1$ (e.g., $2Y = (2\beta+1)(1-\beta)+\lambda \to \lambda$), suggesting that the larger the momentum $\beta$, the smaller the critical batch size $b_{\mathrm{Muon}}^\star$.

**Remark 4.1.** *Previous study* (Sato & Iiduka, 2023) *have considered* $\exists T, \exists b : X/T + Y/b + Z < e_{tot}$ *instead of Eq. (2), and from similar arguments,* $b^\star = \frac{2Y}{\epsilon_{tot} - Z} > \frac{2Y}{\epsilon_{tot}}$ *is obtained. We start from Eq. (2) and obtain* $b^\star = \frac{2Y}{\epsilon}$. *Compared to the former, the relationship between critical batch size* $b^\star$ *and the hyperparameters is clearer. Note that the difference between the two lies in the choice of threshold, i.e.,* $\epsilon := \epsilon_{tot} - Z$ *and both are theoretically correct.*

## 5 Numerical Experiments

We evaluate Muon on three workloads: (i) ResNet-18 on CIFAR-10, (ii) VGG-16 on CIFAR-100, and (iii) Llama3.1 (160M) on the C4 corpus. We first analyze convergence and critical batch size on CIFAR-10 with ResNet-18, then report language–modeling results on C4. Results for VGG-16 on CIFAR-100 are summarized in Appendix F, where we observe the same qualitative trends.

**Experimental Setup** For CIFAR-10 and CIFAR-100 experiments, unless otherwise stated, we tuned the learning rate by grid search at a base batch size of 512 and applied square-root scaling for Muon and AdamW; for Momentum SGD we tried both square-root and linear scaling. Each configuration was run five times with different seeds and we report mean and standard deviation. For C4 dataset [1] we trained Llama3.1 (160M) with sequence length 2048 and batch sizes from 64 to 8192. SFO complexity is measured as *steps × batch size*. Further details of the experimental protocol are in Appendix D.

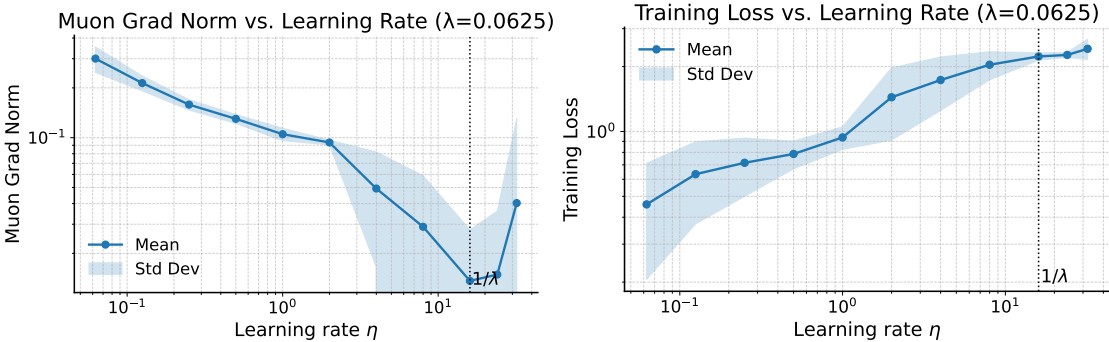

Figure 1: Empirical validation of the stability condition in Proposition 3.2. Final gradient norm (left) and training loss (right) for ResNet-18 on CIFAR-10 with Muon at $\lambda$=0.0625. The dashed line shows $\eta$=1/$\lambda$. Training is most stable near this value.

In the vision workloads, we follow common practice and use a hybrid optimizer (Muon on matrix-shaped parameter blocks and AdamW on the remaining parameters); since our theory analyzes full Muon, we additionally include a controlled full-Muon MLP diagnostic to validate the theory-aligned stopping proxy and the critical-batch predictions (Appendix E).

**Theory-aligned stopping proxy: gradient-norm threshold.** Our theory upper-bounds the average expected full-gradient norm. To better align experiments with this proxy, we additionally report a gradient-norm–based stopping metric. For each run, we track the Frobenius norm of the (mini-batch) gradient, $g_t := \|\nabla f_{\mathcal{S}_t}(W_t)\|_{\mathrm{F}}$, and define a smoothed estimate $\tilde{g}_t$ via an exponential moving average (EMA) over steps.[2] Given a target threshold $\varepsilon_{\mathrm{tot}}$, we define the stopping time $T_\varepsilon(b)$ as the first step such that $\tilde{g}_t \leq \varepsilon_{\mathrm{tot}}$, and report both steps $T_\varepsilon(b)$ and SFO complexity $b \cdot T_\varepsilon(b)$. We emphasize that our original loss/accuracy targets remain useful for practitioner-facing comparisons, while the gradient-norm criterion is introduced specifically to validate the theory-aligned proxy.

---

[1] https://huggingface.co/datasets/allenai/c4

[2] For large-scale workloads, computing the full gradient is impractical; we therefore use the mini-batch gradient norm as an unbiased proxy for the full gradient and report EMA-smoothed curves for stability. In a controlled toy setting (Appendix E), we also compute the full-batch gradient norm to directly match the theoretical quantity.

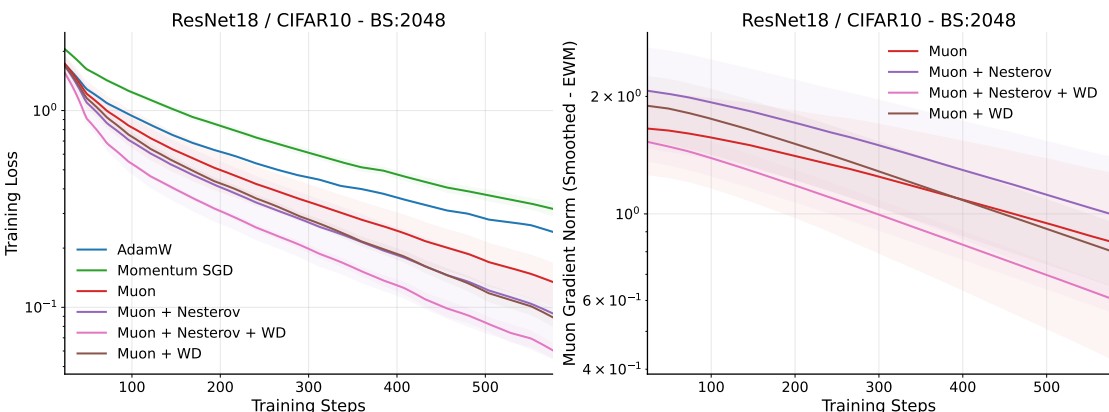

Figure 2: Convergence rate comparison for ResNet-18 on CIFAR-10 with batch size 2048. Training loss (left) and smoothed gradient norm (right) over steps. Muon with Nesterov momentum and weight decay converges the fastest, consistent with the bounds in Table 1.

**Convergence Analysis** We empirically validated the stability condition from Proposition 3.2. Figure 1 shows final gradient norm and training loss for ResNet-18 on CIFAR-10 across learning rates $\eta$ at fixed weight decay $\lambda{=}0.0625$. The vertical dashed line marks the threshold $\eta{=}1/\lambda{=}16.0$. The lowest gradient norm occurs near this threshold; for larger $\eta$ training becomes unstable. The same behavior holds for other values of $\lambda$ (Appendix F).

We next compared the four Muon variants with AdamW and Momentum SGD. Figure 2 shows that Muon with Nesterov momentum and weight decay attains the fastest decrease in both loss and gradient norm.

**Critical Batch Size** We measured the number of steps and SFO needed to reach 90% test accuracy and 95% training accuracy. Figure 3 shows that Muon scales better with batch size than the baselines. SFO is the lowest for Muon over the entire range, and Nesterov shifts the SFO-minimizing batch size to larger values. Momentum SGD requires more steps within the same schedule but eventually reaches comparable accuracy (Appendix F).

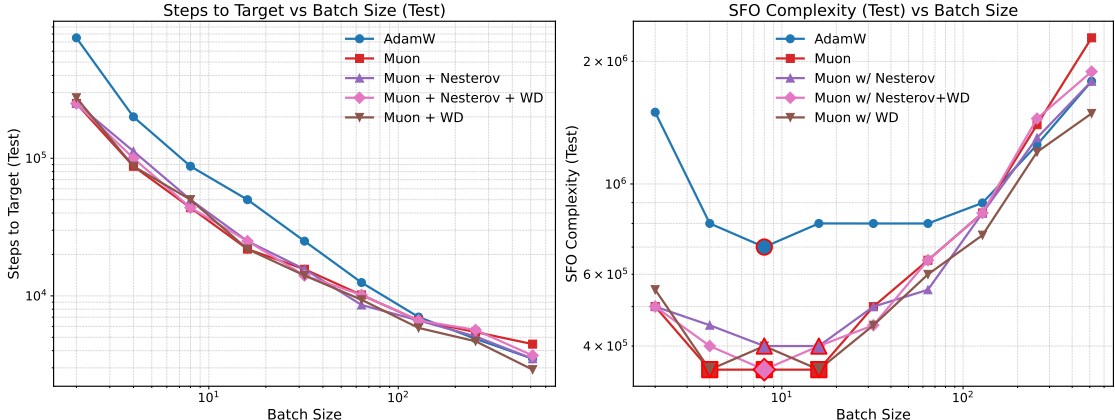

Figure 3: Batch-size scaling and SFO on ResNet-18/CIFAR-10. (Left) Steps to reach 90% test accuracy. (Right) SFO to reach 95% training accuracy. Muon achieves the best efficiency across batch sizes; Nesterov momentum shifts the critical batch size to the right.

**Critical batch size under the gradient-norm stopping proxy (controlled full Muon).** To directly validate the critical-batch phenomenon under the same stopping proxy used in the theory, we run a controlled experiment where Muon is applied to *all* parameters (full Muon) on a small MLP task (Appendix E). We define the empirical SFO as $b \cdot T_\varepsilon(b)$ where $T_\varepsilon(b)$ is the first time the EMA-smoothed gradient norm drops

below $\varepsilon_{\text{tot}}$. We observe a clear U-shaped SFO—batch curve and a well-defined minimizer $b^\star$ (e.g., $b^\star$=32 for $\varepsilon_{\text{tot}}$=0.08), providing evidence that a critical batch size exists when using the theory-aligned gradient-norm criterion.

**Quantitative validation of the** $X/T + Y/b + Z$ **proxy and predicted** $b^\star$**.** We further quantify how well the theoretical decomposition matches observations by fitting

$$\bar{g}(T, b) \;\approx\; \frac{X}{T} + \frac{Y}{b} + Z$$

to the measured average gradient norms collected across multiple $(T, b)$ pairs in the controlled full-Muon MLP setting. A simple linear regression in the features $(1/T, 1/b, 1)$ achieves a high goodness-of-fit (e.g., $R^2$=0.962), supporting that the proxy captures the dominant scaling with $T$ and $b$. From the fitted coefficients, we obtain $\hat{Y}$ and $\hat{Z}$ and compute a *predicted* critical batch size via the theory:

$$b^\star_{\text{pred}} \;=\; \frac{2\hat{Y}}{\varepsilon_{\text{tot}} - \hat{Z}},$$

which matches the empirically SFO-minimizing batch size in the regime $\varepsilon_{\text{tot}} > \hat{Z}$. Moreover, when $\varepsilon_{\text{tot}}$ is not too close to the floor $\hat{Z}$, we observe an approximately linear scaling $b^\star \propto 1/(\varepsilon_{\text{tot}} - \hat{Z})$, consistent with the theoretical prediction.

**Effect of** $\beta$ **on Muon's Critical Batch Size** Theory in Section 4.2 predicts that the critical batch size decreases as $\beta \to 1$. Figure 4 confirms this trend for ResNet-18/CIFAR-10, regardless of weight decay or Nesterov.

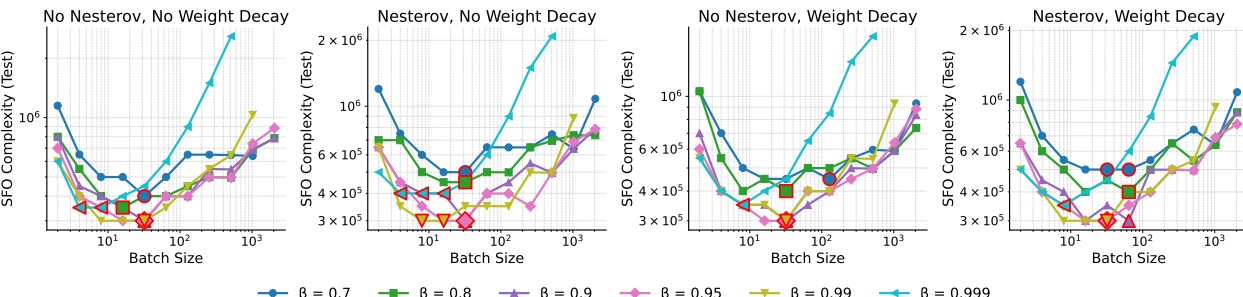

Figure 4: Dependence of SFO and critical batch size on $\beta$ for ResNet-18/CIFAR-10. The critical batch size consistently decreases as $\beta$ increases, in line with Section 4.2.

**Language-Model Workload: C4 on Llama3.1 (160M)** We now test whether the findings hold for large-language-model training. Figure 5 reports final training loss and SFO versus batch size. Muon attains lower loss and lower SFO than AdamW across all batch sizes, and the gap widens at large batches. In this setting, adding Nesterov momentum or weight decay does not yield consistent gains.

**Momentum Sweeps on C4** We varied $\beta$ on C4 to examine the critical batch size. Figure 6 shows that a moderate value, $\beta \approx 0.95$, gives the best loss and SFO. As $\beta$ decreases, the critical batch size increases; as $\beta$ increases toward 1, the critical batch size decreases, but extreme values are suboptimal. These observations are consistent with Section 4.2 and mirror the vision results.

# 6 Related Works

Several studies have investigated the theoretical properties and convergence behavior of Muon. Bernstein & Newhouse (2024) connected Muon to momentum applied in the steepest descent direction under a spectral norm constraint. Li & Hong (2025) provided a pioneering analysis assuming Frobenius norm Lipschitz smoothness. Pethick et al. (2025) studied Muon in the context of optimization methods that use linear

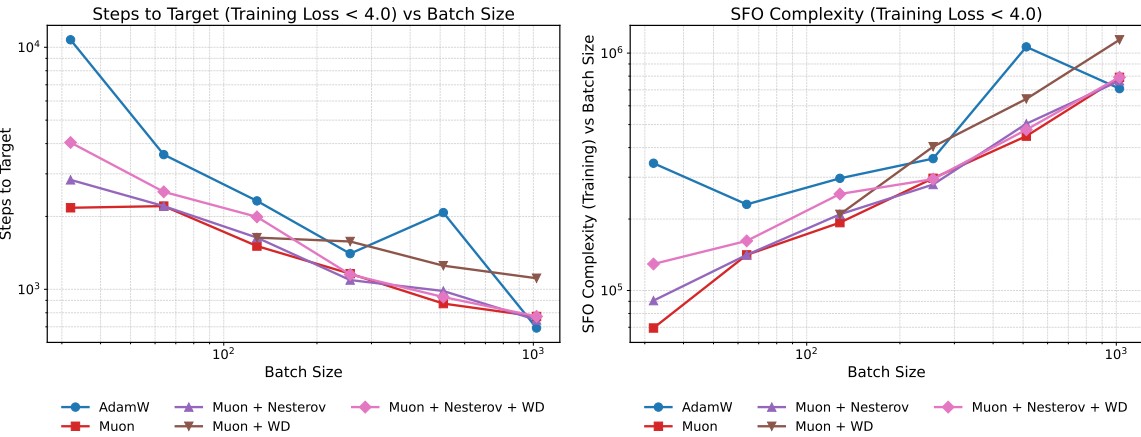

Figure 5: Batch-size scaling on `C4` with `Llama3.1 (160M)`. Steps to reach the target training loss (left) and SFO complexity (right) versus batch size. Muon outperforms AdamW in terms of both the number of steps required to reach the target loss and the SFO complexity in almost all cases. Nesterov momentum and weight decay provide little additional benefit for this workload.

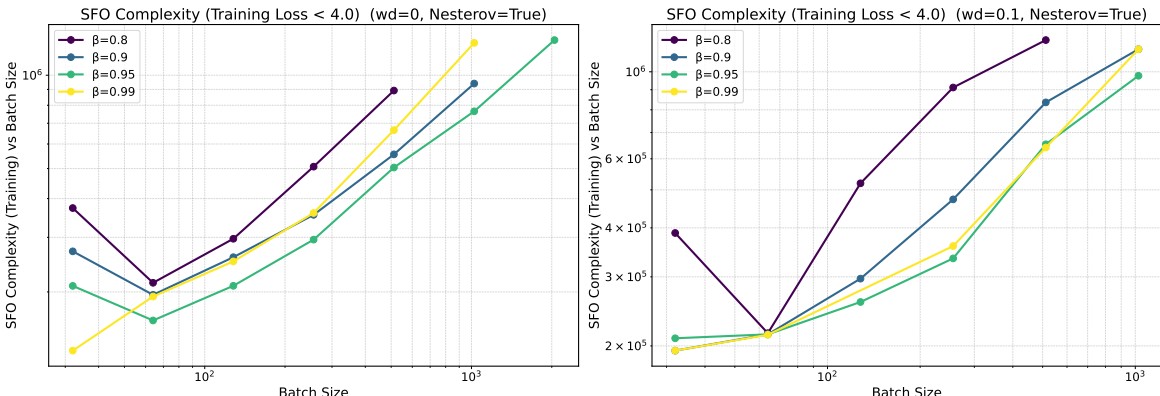

Figure 6: Effect of momentum $\beta$ on `C4` with `Llama3.1 (160M)`. Loss and SFO across batch sizes for different $\beta$. The best trade-off occurs near $\beta=0.95$. The critical batch size decreases with larger $\beta$.

minimization oracles for norm balls, and established a convergence rate. Connections to other optimizers have also been explored. Shah et al. showed that Shampoo and SOAP reduce to Muon under simplifying assumptions (AI et al., 2025). Kovalev (2025) proposed and analyzed a stochastic non-Euclidean trust-region method that includes Muon as a special case. Similarly, An et al. (2025) proposed an adaptive structured gradient optimization algorithm that matches Muon in its momentum-free variant. Other studies have explored specific properties and extensions of Muon. Petrov et al. (2025) proposed and analyzed a zeroth-order version . Chen et al. demonstrated Muon's compatibility with the Lion-$\mathcal{K}$ algorithm (Chen et al., 2024) and showed that Muon with weight decay implicitly solves an optimization problem with a spectral norm constraint (Chen et al., 2025). Shen et al. (2025) presented a comprehensive analysis of Muon's convergence rate in comparison to gradient descent. Lau et al. (2025) introduced PolarGrad, a unifying framework for matrix-aware preconditioned methods including Muon, and established convergence rates. The core concepts of gradient orthogonalization and dualization, which are central to Muon, were introduced in the foundational works by Carlson et al. (2015) and Flynn (2017). The convergence rates of major related works are summarized in Table 3. Our novelty lies in analyzing the upper bound on the gradient norm of Muon with Nesterov momentum and weight decay.

The framework for assumptions and proofs presented by Shen et al. (2025) is the most similar to our own. The crucial difference between our study and all previous studies, including this one, is that we also consider

most common variant of Muon, which incorporates Nesterov momentum and weight decay. Technically, the technique of transforming $\langle C_t, O_t \rangle_{\mathrm{F}}$ using the dual norm and inverse triangle inequality follows the proof of Pethick et al. (2025) (see proof of Theorem B.1 in Appendix B).

Table 3: Comparison of convergence rates in related works. $\|\cdot\|_\star$ denotes an arbitrary norm, and $\|\cdot\|_*$ denotes the nuclear norm. Each result has been rewritten to conform to our notation. $S(W_t)$ is the KKT score function defined as $S(W) := \|\nabla f(W)\|_* + \lambda \langle W, \nabla f(W) \rangle$.

| Related work | Measure | Convergence Rate | Nesterov momentum | Weight decay |
|---|---|---|---|---|
| Pethick et al. (2025) | $\mathbb{E}\left[\|\nabla f(W_T)\|_\star\right]$ | $\mathcal{O}\left(\frac{1}{T} + \eta\right)$ | ✗ | ✗ |
| Li & Hong (2025) | $\frac{1}{T}\sum_{t=0}^{T-1}\mathbb{E}\left[\|\nabla f(W_t)\|_{\mathrm{F}}\right]$ | $\mathcal{O}\left(\frac{1}{T} + \frac{1}{\sqrt{b}} + n\right)$ | ✗ | ✗ |
| Kovalev (2025) | $\min_{0\le t\le T-1}\mathbb{E}\left[\|\nabla f(W_t)\|_*\right]$ | $\mathcal{O}\left(\frac{1}{T} + \eta + \sqrt{\beta}\right)$ | ✗ | ✗ |
| Shen et al. (2025) | $\frac{1}{T}\sum_{t=0}^{T-1}\mathbb{E}\left[\|\nabla f(W_t)\|_*\right]$ | $\mathcal{O}\left(\frac{1}{T} + \frac{1}{\sqrt{b}} + r\right)$ | ✗ | ✗ |
| Chen et al. (2025) | $\frac{1}{T}\sum_{t=0}^{T-1}\mathbb{E}\left[S(W_t)\right]$ | $\mathcal{O}\left(\frac{1}{T} + \frac{1}{\sqrt{b}} + n\right)$ | ✓ | ✓ |
| Lau et al. (2025) | $\frac{1}{T}\min_{0\le t\le T-1}\mathbb{E}\left[\|\nabla f(W_t)\|_{\mathrm{F}}\right]$ | $\mathcal{O}\left(\frac{1}{T} + \eta + \sqrt{r}\right)$ | ✗ | ✗ |
| Ours (Theorem 3.2(ii)) | $\frac{1}{T}\sum_{t=0}^{T-1}\mathbb{E}\left[\|\nabla f(W_t)\|_{\mathrm{F}}\right]$ | $\mathcal{O}\left(\frac{1}{T} + \frac{1}{\sqrt{b}} + n\right)$ | ✓ | ✓ |

# 7 Conclusion

Through a comprehensive theoretical analysis of the Muon optimizer, we established convergence guarantees for four practical configurations (with and without Nesterov momentum and with and without weight decay). Our primary theoretical contribution is demonstrating the crucial role of weight decay. We proved that it enforces a strict decrease in parameter and gradient norms, a clear advantage over the standard Muon configuration. This theoretical insight, along with the necessary condition relating the learning rate and weight decay coefficient, was empirically validated by our experimental results. Additionally, we derived the critical batch size for Muon, revealing its dependence on fundamental hyperparameters such as momentum and weight decay. Collectively, our findings provide both a deeper theoretical understanding of Muon and actionable guidance for practitioners aiming to leverage this promising optimizer in large-scale settings.

Finally, we note a limitation regarding our single-matrix analysis. In deep neural networks, gradient statistics vary across layers. However, recent work by Gray et al. (2024) shows that gradient noise scales of different layers are highly correlated. This suggests that while our theory does not predict exact aggregate values, it captures the fundamental functional dependencies of the critical batch size on hyperparameters.

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

## A   Tools for Proof of All Theorems

The results presented in this section are not new and are given simply for reference and completeness.

**Lemma A.1.** *Suppose Assumption 2.2(ii) hold for all $t \in \mathbb{N}$; then,*

$$\mathbb{E}_{\boldsymbol{\xi}_t} \left[ \|\nabla f_{\mathcal{S}_t}(W_t) - \nabla f(W_t)\|_{\mathrm{F}}^2 \right] \leq \frac{C^2}{b}.$$

*Proof.* Assumption 2.2(ii) guarantee that

$$
\begin{aligned}
\mathbb{E}_{\boldsymbol{\xi}_t} \left[ \|\nabla f_{\mathcal{S}_t}(W_t) - \nabla f(W_t)\|_{\mathrm{F}}^2 \right] &= \mathbb{E}_{\boldsymbol{\xi}_t} \left[ \left\| \frac{1}{b} \sum_{i=1}^{b} \mathsf{G}_{\xi_{t,i}}(W_t) - \nabla f(W_t) \right\|_{\mathrm{F}}^2 \right] \\
&= \mathbb{E}_{\boldsymbol{\xi}_t} \left[ \left\| \frac{1}{b} \sum_{i=1}^{b} \mathsf{G}_{\xi_{t,i}}(W_t) - \frac{1}{b} \sum_{i=1}^{b} \nabla f(W_t) \right\|_{\mathrm{F}}^2 \right] \\
&= \mathbb{E}_{\boldsymbol{\xi}_t} \left[ \left\| \frac{1}{b} \sum_{i=1}^{b} \left( \mathsf{G}_{\xi_{t,i}}(W_t) - \nabla f(W_t) \right) \right\|_{\mathrm{F}}^2 \right] \\
&= \frac{1}{b^2} \mathbb{E}_{\boldsymbol{\xi}_t} \left[ \left\| \sum_{i=1}^{b} \left( \mathsf{G}_{\xi_{t,i}}(W_t) - \nabla f(W_t) \right) \right\|_{\mathrm{F}}^2 \right] \\
&= \frac{1}{b^2} \mathbb{E}_{\boldsymbol{\xi}_t} \left[ \sum_{i=1}^{b} \left\| \mathsf{G}_{\xi_{t,i}}(W_t) - \nabla f(W_t) \right\|_{\mathrm{F}}^2 \right] \\
&\leq \frac{\sigma^2}{b}.
\end{aligned}
$$

This completes the proof. $\qquad\square$

The following lemma was established by (Mokhtari et al., 2020). In their setting, the algorithm without momentum corresponds to the case in which $\beta_1 = 1$, whereas in ours it corresponds to $\beta_1 = 0$. As a result, the statements may appear slightly different.

**Lemma A.2.** *Suppose Assumptions 2.1 and 2.2 hold. Then for all $t \in \mathbb{N}$,*

$$\sum_{t=0}^{T-1} \mathbb{E} \left[ \|M_t - \nabla f(W_t)\|_{\mathrm{F}}^2 \right] \leq \frac{2}{1-\beta} \|M_0 - \nabla f(W_0)\|_{\mathrm{F}}^2 + \frac{2(1-\beta)\sigma^2}{b} T + \frac{4L^2\eta^2 n}{(1-\beta)^2} T,$$

*and*

$$\sum_{t=0}^{T-1} \mathbb{E} \left[ \|M_t - \nabla f(W_t)\|_{\mathrm{F}} \right] \leq \frac{2\sqrt{2}}{1-\beta} \|M_0 - \nabla f(W_0)\|_{\mathrm{F}} + \sqrt{\frac{2(1-\beta)\sigma^2}{b}} T + \frac{2L\eta\sqrt{n}}{1-\beta} T.$$

*Proof.* From the definition of $M_t$,

$$
\begin{aligned}
\|M_t - \nabla f(W_t)\|_{\mathrm{F}}^2 &= \|\beta M_{t-1} + (1-\beta)\nabla f_{\mathcal{S}_t}(W_t) - \nabla f(W_t)\|_{\mathrm{F}}^2 \\
&= \|\beta(M_{t-1} - \nabla f(W_{t-1})) + \beta(\nabla f(W_{t-1}) - \nabla f(W_t)) + (1-\beta)(\nabla f_{\mathcal{S}_t}(W_t) - \nabla f(W_t))\|_{\mathrm{F}}^2 \\
&= \beta^2 \|M_{t-1} - \nabla f(W_{t-1})\|_{\mathrm{F}}^2 + \beta^2 \|\nabla f(W_{t-1}) - \nabla f(W_t)\|_{\mathrm{F}}^2 + (1-\beta)^2 \|\nabla f_{\mathcal{S}_t}(W_t) - \nabla f(W_t)\|_{\mathrm{F}}^2 \\
&\quad + 2\beta^2 \langle M_{t-1} - \nabla f(W_{t-1}), \nabla f(W_{t-1}) - \nabla f(W_t) \rangle_{\mathrm{F}} \\
&\quad + 2\beta(1-\beta) \langle M_{t-1} - \nabla f(W_{t-1}), \nabla f_{\mathcal{S}_t}(W_t) - \nabla f(W_t) \rangle_{\mathrm{F}} \\
&\quad + 2\beta(1-\beta) \langle \nabla f(W_{t-1}) - \nabla f(W_t), \nabla f_{\mathcal{S}_t}(W_t) - \nabla f(W_t) \rangle_{\mathrm{F}}
\end{aligned}
$$

Therefore, by taking the expectation,

$$\mathbb{E}\left[\|M_t - \nabla f(W_t)\|_{\mathrm{F}}^2\right] = \beta^2 \mathbb{E}\left[\|M_{t-1} - \nabla f(W_{t-1})\|_{\mathrm{F}}^2\right] + \beta^2 \mathbb{E}\left[\|\nabla f(W_{t-1}) - \nabla f(W_t)\|_{\mathrm{F}}^2\right]$$
$$+ (1-\beta)^2 \mathbb{E}\left[\|\nabla f_{\mathcal{S}_t}(W_t) - \nabla f(W_t)\|_{\mathrm{F}}^2\right] + 2\beta^2 \mathbb{E}\left[\langle M_{t-1} - \nabla f(W_{t-1}), \nabla f(W_{t-1}) - \nabla f(W_t)\rangle_{\mathrm{F}}\right].$$

Here, by the Peter-Paul inequality, for all $\epsilon > 0$, we have

$$\langle M_{t-1} - \nabla f(W_{t-1}), \nabla f(W_{t-1}) - \nabla f(W_t)\rangle_{\mathrm{F}} \leq \frac{\epsilon}{2}\|M_{t-1} - \nabla f(W_{t-1})\|_{\mathrm{F}}^2 + \frac{1}{2\epsilon}\|\nabla f(W_{t-1}) - \nabla f(W_t)\|_{\mathrm{F}}^2.$$

Therefore, we obtain

$$\mathbb{E}\left[\|M_t - \nabla f(W_t)\|_{\mathrm{F}}^2\right] = \beta^2(1+\epsilon)\mathbb{E}\left[\|M_{t-1} - \nabla f(W_{t-1})\|_{\mathrm{F}}^2\right] + \beta^2\left(1 + \frac{1}{\epsilon}\right)\mathbb{E}\left[\|\nabla f(W_{t-1}) - \nabla f(W_t)\|_{\mathrm{F}}^2\right]$$
$$+ (1-\beta)^2 \mathbb{E}\left[\|\nabla f_{\mathcal{S}_t}(W_t) - \nabla f(W_t)\|_{\mathrm{F}}^2\right].$$

In addition, from Assumption 2.1,

$$\|\nabla f(W_{t-1}) - \nabla f(W_t)\|_{\mathrm{F}}^2 \leq L^2\|W_{t-1} - W_t\|_{\mathrm{F}}^2 = L^2\eta^2\|O_t\|_{\mathrm{F}}^2 \leq L^2\eta^2 n.$$

Hence, from Lemma A.1,

$$\mathbb{E}\left[\|M_t - \nabla f(W_t)\|_{\mathrm{F}}^2\right] \leq \beta^2(1+\epsilon)\mathbb{E}\left[\|M_{t-1} - \nabla f(W_{t-1})\|_{\mathrm{F}}^2\right] + \beta^2\left(1 + \frac{1}{\epsilon}\right)L^2\eta^2 n + \frac{(1-\beta)^2\sigma^2}{b}.$$

Then, letting $\epsilon := \frac{1-\beta}{2}$, we have

$$\mathbb{E}\left[\|M_t - \nabla f(W_t)\|_{\mathrm{F}}^2\right] \leq \frac{\beta^2(3-\beta)}{2}\mathbb{E}\left[\|M_{t-1} - \nabla f(W_{t-1})\|_{\mathrm{F}}^2\right] + \frac{\beta^2(3-\beta)}{1-\beta}L^2\eta^2 n + \frac{(1-\beta)^2\sigma^2}{b}$$
$$\leq \frac{1+\beta}{2}\mathbb{E}\left[\|M_{t-1} - \nabla f(W_{t-1})\|_{\mathrm{F}}^2\right] + \frac{2}{1-\beta}L^2\eta^2 n + \frac{(1-\beta)^2\sigma^2}{b}$$
$$\leq \left(\frac{1+\beta}{2}\right)^t \|M_0 - \nabla f(W_0)\|_{\mathrm{F}}^2 + \left\{\frac{2L^2\eta^2 n}{1-\beta} + \frac{(1-\beta)^2\sigma^2}{b}\right\}\sum_{k=0}^{t-1}\left(\frac{1+\beta}{2}\right)^k$$
$$\leq \left(\frac{1+\beta}{2}\right)^t \|M_0 - \nabla f(W_0)\|_{\mathrm{F}}^2 + \left\{\frac{2L^2\eta^2 n}{1-\beta} + \frac{(1-\beta)^2\sigma^2}{b}\right\}\frac{2}{1-\beta}$$
$$= \left(\frac{1+\beta}{2}\right)^t \|M_0 - \nabla f(W_0)\|_{\mathrm{F}}^2 + \frac{4L^2\eta^2 n}{(1-\beta)^2} + \frac{2(1-\beta)\sigma^2}{b}.$$

Therefore, summing over $t$, we have

$$\sum_{t=0}^{T-1}\mathbb{E}\left[\|M_t - \nabla f(W_t)\|_{\mathrm{F}}^2\right] \leq \frac{2}{1-\beta}\|M_0 - \nabla f(W_0)\|_{\mathrm{F}}^2 + \frac{2(1-\beta)\sigma^2}{b}T + \frac{4L^2\eta^2 n}{(1-\beta)^2}T.$$

Finally, from the properties of variance and expectation,

$$\mathbb{E}\left[\|M_t - \nabla f(W_t)\|_{\mathrm{F}}\right] \leq \sqrt{\mathbb{E}\left[\|M_t - \nabla f(W_t)\|_{\mathrm{F}}^2\right]}$$
$$\leq \sqrt{\left(\frac{1+\beta}{2}\right)^t}\|M_0 - \nabla f(W_0)\|_{\mathrm{F}} + \sqrt{\frac{2(1-\beta)\sigma^2}{b}} + \frac{2L\eta\sqrt{n}}{(1-\beta)}.$$

Hence, we have

$$\sum_{t=0}^{T-1}\mathbb{E}\left[\|M_t - \nabla f(W_t)\|_{\mathrm{F}}\right] \leq \frac{\sqrt{2}}{\sqrt{2} - \sqrt{1+\beta}}\|M_0 - \nabla f(W_0)\|_{\mathrm{F}} + \sqrt{\frac{2(1-\beta)\sigma^2}{b}}T + \frac{2L\eta\sqrt{n}}{1-\beta}T$$
$$\leq \frac{2\sqrt{2}}{1-\beta}\|M_0 - \nabla f(W_0)\|_{\mathrm{F}} + \sqrt{\frac{2(1-\beta)\sigma^2}{b}}T + \frac{2L\eta\sqrt{n}}{1-\beta}T$$

This completes the proof. □

# B  Proof of Theorems for Muon without weight decay

**Theorem B.1** (Auxiliary Theorem for Muon without weight decay). *Suppose Assumptions 2.1 and 2.2 hold. Then for all $t \in \mathbb{N}$,*

$$\sum_{t=0}^{T-1} \mathbb{E}\left[\|\nabla f(W_t)\|_{\mathrm{F}}\right] \leq \frac{f(W_0) - f(W_T)}{\eta} + \frac{1}{2}\sum_{t=0}^{T-1}\mathbb{E}\left[\|\nabla f(W_t) - C_t\|_{\mathrm{F}}^2\right] + \sqrt{r}\sum_{t=0}^{T-1}\mathbb{E}\left[\|\nabla f(W_t) - C_t\|_{\mathrm{F}}\right] + \frac{1 + L\eta}{2}nT,$$

*where* $\mathrm{rank}(C_t - \nabla f(W_t)) =: r_t \leq \max_{0 \leq t \leq T-1} r_t =: r.$

*Proof.* From Assumption 2.1,

$$f(W_{t+1}) \leq f(W_t) + \langle \nabla f(W_t), W_{t+1} - W_t \rangle_{\mathrm{F}} + \frac{L}{2}\|W_{t+1} - W_t\|_{\mathrm{F}}^2$$

$$= f(W_t) - \eta\langle \nabla f(W_t), O_t \rangle_{\mathrm{F}} + \frac{L}{2}\eta^2\|O_t\|_{\mathrm{F}}^2$$

$$\leq (W_t) - \eta\langle C_t, O_t \rangle_{\mathrm{F}} - \eta\langle \nabla f(W_t) - C_t, O_t \rangle_{\mathrm{F}} + \frac{L\eta^2 n}{2}.$$

Using the definition $O_t := \underset{O \in \{O \in \mathbb{R}^{m \times n}: O^\top O = I_n\}}{\mathrm{argmin}} \|O - C_t\|_{\mathrm{F}}$, we obtain $O_t := \underset{O \in \{O \in \mathbb{R}^{m \times n}: O^\top O = I_n\}}{\mathrm{argmax}} \langle C_t, O \rangle_{\mathrm{F}}.$
Then,

$$\langle C_t, O_t \rangle_{\mathrm{F}} = \max_{O:O^\top O = I_n} \langle C_t, O \rangle_{\mathrm{F}} = \max_{O:\|O\|_2 \leq 1} \langle C_t, O \rangle_{\mathrm{F}} =: \|C_t\|_*,$$

where $\|\cdot\|_*$ denotes the dual norm. Applying the reverse triangle inequality and the relation $\|A\|_{\mathrm{F}} \leq \|A\|_* \leq \sqrt{\mathrm{rank}(A)}\|A\|_{\mathrm{F}}$, we have

$$-\langle C_t, O_t \rangle_{\mathrm{F}} = -\|C_t\|_*$$
$$= -\|C_t - \nabla f(W_t) + \nabla f(W_t)\|_*$$
$$\leq \|C_t - \nabla f(W_t)\|_* - \|\nabla f(W_t)\|_*$$
$$\leq \sqrt{\mathrm{rank}(C_t - \nabla f(W_t))}\|C_t - \nabla f(W_t)\|_{\mathrm{F}} - \|\nabla f(W_t)\|_{\mathrm{F}}$$
$$\leq \sqrt{r}\|C_t - \nabla f(W_t)\|_{\mathrm{F}} - \|\nabla f(W_t)\|_{\mathrm{F}}, \tag{6}$$

where $\mathrm{rank}(C_t - \nabla f(W_t)) =: r_t \leq \max_{0 \leq t \leq T-1} r_t =: r \ (\leq n)$. In addition, we have

$$-\langle \nabla f(W_t) - C_t, O_t \rangle_{\mathrm{F}} = \frac{1}{2}\left(\|\nabla f(W_t) - C_t\|_{\mathrm{F}}^2 + \|O_t\|_{\mathrm{F}}^2 - \|\nabla f(W_t) - C_t + O_t\|_{\mathrm{F}}^2\right)$$

$$\leq \frac{1}{2}\|\nabla f(W_t) - C_t\|_{\mathrm{F}}^2 + \frac{n}{2}.$$

Therefore,

$$f(W_{t+1}) \leq f(W_t) + \eta\sqrt{r}\|C_t - \nabla f(W_t)\|_{\mathrm{F}} - \eta\|\nabla f(W_t)\|_{\mathrm{F}} + \frac{\eta}{2}\|\nabla f(W_t) - C_t\|_{\mathrm{F}}^2 + \frac{1 + L\eta}{2}\eta n.$$

By rearranging the terms and taking expectation, we obtain

$$\mathbb{E}\left[\|\nabla f(W_t)\|_{\mathrm{F}}\right] \leq \frac{f(W_t) - f(W_{t+1})}{\eta} + \frac{1}{2}\mathbb{E}\left[\|\nabla f(W_t) - C_t\|_{\mathrm{F}}^2\right] + \sqrt{r}\mathbb{E}\left[\|\nabla f(W_t) - C_t\|_{\mathrm{F}}\right] + \frac{1 + L\eta}{2}n.$$

This completes the proof. $\qquad\square$

## B.1 Proof of Theorem 3.1(i)

*Proof.* From $C_t := M_t$, together with Lemmas A.1 and A.2 and Theorem B.1, we find that

$$\sum_{t=0}^{T-1} \mathbb{E}\left[\|\nabla f(W_t)\|_{\mathrm{F}}\right] \leq \frac{f(W_0) - f(W_T)}{\eta} + \frac{1}{2}\sum_{t=0}^{T-1} \mathbb{E}\left[\|\nabla f(W_t) - M_t\|_{\mathrm{F}}^2\right] + \sqrt{r}\sum_{t=0}^{T-1} \mathbb{E}\left[\|\nabla f(W_t) - M_t\|_{\mathrm{F}}\right] + \frac{1 + L\eta}{2}nT$$

$$\leq \frac{f(W_0) - f(W_T)}{\eta} + \frac{1}{2}\left\{\frac{2}{1-\beta}\|M_0 - \nabla f(W_0)\|_{\mathrm{F}}^2 + \frac{2(1-\beta)\sigma^2}{b}T + \frac{4L^2\eta^2 n}{(1-\beta)^2}T\right\}$$

$$+ \sqrt{r}\left\{\frac{2\sqrt{2}}{1-\beta}\|M_0 - \nabla f(W_0)\|_{\mathrm{F}} + \sqrt{\frac{2(1-\beta)\sigma^2}{b}T} + \frac{2L\eta\sqrt{n}}{1-\beta}T\right\} + \frac{1 + L\eta}{2}nT.$$

By taking the average over $t = 0, \ldots, T-1$ and applying expectation, we obtain

$$\frac{1}{T}\sum_{t=0}^{T-1} \mathbb{E}\left[\|\nabla f(W_t)\|_{\mathrm{F}}\right] \leq \frac{f(W_0) - f(W_T)}{\eta T} + \frac{\|M_0 - \nabla f(W_0)\|_{\mathrm{F}}^2}{(1-\beta)T} + \frac{2\sqrt{2r}\|M_0 - \nabla f(W_0)\|_{\mathrm{F}}}{(1-\beta)T}$$

$$+ \frac{(1-\beta)\sigma^2}{b} + \sqrt{\frac{2(1-\beta)r\sigma^2}{b}} + \frac{2L^2\eta^2 n}{(1-\beta)^2} + \frac{2L\eta\sqrt{nr}}{1-\beta} + \frac{1 + L\eta}{2}n$$

$$= \mathcal{O}\left(\frac{1}{T} + \frac{1-\beta}{b} + n\right).$$

This completes the proof. □

## B.2 Proof of Theorem 3.1(ii)

*Proof.* From the definition of $C_t$, we have

$$\|C_t - \nabla f(W_t)\|_{\mathrm{F}} = \|\beta M_t + (1-\beta)\nabla f_{\mathcal{S}_t}(W_t) - \nabla f(W_t)\|_{\mathrm{F}}$$
$$= \|\beta(M_t - \nabla f(W_t)) + (1-\beta)(\nabla f_{\mathcal{S}_t}(W_t) - \nabla f(W_t))\|_{\mathrm{F}}$$
$$\leq \beta\|M_t - \nabla f(W_t)\|_{\mathrm{F}} + (1-\beta)\|\nabla f_{\mathcal{S}_t}(W_t) - \nabla f(W_t)\|_{\mathrm{F}}, \tag{7}$$

and

$$\|C_t - \nabla f(W_t)\|_{\mathrm{F}}^2 \leq \beta\|M_t - \nabla f(W_t)\|_{\mathrm{F}}^2 + (1-\beta)\|\nabla f_{\mathcal{S}_t}(W_t) - \nabla f(W_t)\|_{\mathrm{F}}^2. \tag{8}$$

According to Theorem B.1, we find that

$$\sum_{t=0}^{T-1} \mathbb{E}\left[\|\nabla f(W_t)\|_{\mathrm{F}}\right] \leq \frac{f(W_0) - f(W_T)}{\eta} + \frac{\beta}{2}\sum_{t=0}^{T-1} \mathbb{E}\left[\|\nabla f(W_t) - M_t\|_{\mathrm{F}}^2\right] + \frac{1-\beta}{2}\sum_{t=0}^{T-1} \mathbb{E}\left[\|\nabla f_{\mathcal{S}_t}(W_t) - \nabla f(W_t)\|_{\mathrm{F}}^2\right]$$

$$+ \beta\sqrt{r}\sum_{t=0}^{T-1} \mathbb{E}\left[\|\nabla f(W_t) - M_t\|_{\mathrm{F}}\right] + (1-\beta)\sqrt{r}\sum_{t=0}^{T-1} \mathbb{E}\left[\|\nabla f_{\mathcal{S}_t}(W_t) - \nabla f(W_t)\|_{\mathrm{F}}\right] + \frac{1 + L\eta}{2}nT.$$

From Lemmas A.1 and A.2, we have

$$\sum_{t=0}^{T-1} \mathbb{E}\left[\|\nabla f(W_t)\|_{\mathrm{F}}\right]$$

$$\leq \frac{f(W_0) - f(W_T)}{\eta} + \frac{\beta}{2}\left\{\frac{2}{1-\beta}\|M_0 - \nabla f(W_0)\|_{\mathrm{F}}^2 + \frac{2(1-\beta)\sigma^2}{b}T + \frac{4L^2\eta^2 n}{(1-\beta)^2}T\right\}$$

$$+ \frac{1-\beta}{2}\cdot\frac{\sigma^2}{b}T + \beta\sqrt{r}\left\{\frac{2\sqrt{2}}{1-\beta}\|M_0 - \nabla f(W_0)\|_{\mathrm{F}} + \sqrt{\frac{2(1-\beta)\sigma^2}{b}T} + \frac{2L\eta\sqrt{n}}{1-\beta}T\right\}$$

$$+ (1-\beta)\sqrt{r}\cdot\sqrt{\frac{\sigma^2}{b}T} + \frac{1 + L\eta}{2}nT.$$

By taking the average over $t = 0, \ldots, T - 1$ and applying expectation, we obtain

$$\frac{1}{T} \sum_{t=0}^{T-1} \mathbb{E}\left[\|\nabla f(W_t)\|_2\right] \leq \frac{f(W_0) - f(W_T)}{\eta T} + \frac{\beta\|M_0 - \nabla f(W_0)\|_{\mathrm{F}}^2}{(1-\beta)T} + \frac{2\beta\sqrt{2r}\|M_0 - \nabla f(W_0)\|_{\mathrm{F}}}{(1-\beta)T}$$

$$+ \frac{(2\beta+1)(1-\beta)}{2} \frac{\sigma^2}{b} + (\beta\sqrt{2(1-\beta)} + (1-\beta))\sqrt{\frac{r\sigma^2}{b}}$$

$$+ \frac{2L^2\eta^2\beta n}{(1-\beta)^2} + \frac{2L\eta\sqrt{nr}\beta}{1-\beta} + \frac{1+L\eta}{2}n$$

$$= \mathcal{O}\left(\frac{1}{T} + \frac{(2\beta+1)(1-\beta)}{2} \cdot \frac{1}{b} + n\right)$$

This completes the proof. $\qquad\square$

## C  Proof of Theorems for Muon with weight decay

### C.1  Proof of Proposition 3.1

*Proof.* From the definition of $W_t$ and the condition $\eta \leq \frac{1}{\lambda}$, we have

$$\|W_t\|_{\mathrm{F}} = \|(1-\eta\lambda)W_{t-1} - \eta O_t\|_{\mathrm{F}}$$

$$\leq (1-\eta\lambda)\|W_{t-1}\|_{\mathrm{F}} + \eta\sqrt{n}$$

$$\leq (1-\eta\lambda)^t\|W_0\|_{\mathrm{F}} + \eta\sqrt{n}\sum_{k=0}^{t-1}(1-\eta\lambda)^k$$

$$\leq (1-\eta\lambda)^t\|W_0\|_{\mathrm{F}} + \frac{\sqrt{n}}{\lambda}$$

$$\leq (1-\eta\lambda)^t\|W_0\|_{\mathrm{F}} + \frac{\sqrt{n}}{\lambda}.$$

This completes the proof. $\qquad\square$

### C.2  Proof of Proposition 3.2

*Proof.* According to Assumption 2.1,

$$\|\nabla f(W_t) - \nabla f(W^\star)\|_{\mathrm{F}} \leq L\|W_t - W^\star\|_{\mathrm{F}}.$$

Therefore, from Proposition 3.1 and the fact that $\nabla f(W^\star) = \mathbf{0}$, we have

$$\|\nabla f(W_t)\|_{\mathrm{F}} \leq L\|W_t - W^\star\|_{\mathrm{F}}$$

$$\leq L\|W_t\|_{\mathrm{F}} + L\|W^\star\|_{\mathrm{F}}$$

$$\leq L(1-\eta\lambda)^t\|W_0\|_{\mathrm{F}} + \frac{L\sqrt{n}}{\lambda} + L\|W^\star\|_{\mathrm{F}}.$$

This completes the proof. $\qquad\square$

**Lemma C.1.** *Suppose Assumptions 2.1 and 2.2 hold and $\eta \leq \frac{1}{\lambda}$. Then, for all $t \in \mathbb{N}$,*

$$\sum_{t=0}^{T-1} \mathbb{E}\left[\|\nabla f_{\mathcal{S}_t}(W_t)\|_{\mathrm{F}}^2\right] \leq \left(\frac{\sigma^2}{b} + D_0^2\right)T \quad and \quad \sum_{t=0}^{T-1} \mathbb{E}\left[\|M_t\|_{\mathrm{F}}^2\right] \leq \left(\frac{\sigma^2}{b} + D_0^2\right)T,$$

*where $D_0 := L\left(\|W_0\|_{\mathrm{F}} + \frac{\sqrt{n}}{\lambda} + \|W^\star\|_{\mathrm{F}}\right)$.*

*Proof.* According to Assumption 2.2(i),

$$\mathbb{E}\left[\|\nabla f_{\mathcal{S}_t}(W_t) - \nabla f(W_t)\|_{\mathrm{F}}^2\right] = \mathbb{E}\left[\|\nabla f_{\mathcal{S}_t}(W_t)\|_{\mathrm{F}}^2\right] - 2\mathbb{E}\left[\langle\nabla f_{\mathcal{S}_t}(W_t), \nabla f(W_t)\rangle_{\mathrm{F}}\right] + \|\nabla f(W_t)\|_{\mathrm{F}}^2$$
$$= \mathbb{E}\left[\|\nabla f_{\mathcal{S}_t}(W_t)\|_{\mathrm{F}}^2\right] - \|\nabla f(W_t)\|_{\mathrm{F}}^2.$$

Then, from Lemma A.1 and Proposition 3.2, we have

$$\mathbb{E}\left[\|\nabla f_{\mathcal{S}_t}(W_t)\|_{\mathrm{F}}^2\right] \leq \frac{\sigma^2}{b} + \|\nabla f(W_t)\|_2^2$$
$$\leq \frac{\sigma^2}{b} + D_0^2.$$

Hence, we have

$$\sum_{t=0}^{T-1} \mathbb{E}\left[\|\nabla f_{\mathcal{S}_t}(W_t)\|_{\mathrm{F}}^2\right] \leq \left(\frac{\sigma^2}{b} + D_0^2\right) T.$$

Next, from the definition of $M_t$,

$$\mathbb{E}\left[\|M_t\|_{\mathrm{F}}^2\right] = \mathbb{E}\left[\|\beta M_{t-1} + (1-\beta)\nabla f_{\mathcal{S}_t}(W_t)\|_{\mathrm{F}}^2\right]$$
$$\leq \beta\mathbb{E}\left[\|M_{t-1}\|_{\mathrm{F}}^2\right] + (1-\beta)\mathbb{E}\left[\|\nabla f_{\mathcal{S}_t}(W_t)\|_{\mathrm{F}}^2\right]$$
$$\leq \beta\mathbb{E}\left[\|M_{t-1}\|_{\mathrm{F}}^2\right] + (1-\beta)\left(\frac{\sigma^2}{b} + D_t^2\right)$$
$$\leq \beta^{t+1}\|M_{-1}\|_{\mathrm{F}}^2 + (1-\beta)\left(\frac{\sigma^2}{b} + D_0^2\right)\sum_{k=0}^{t}\beta^k$$
$$\leq \left(\frac{\sigma^2}{b} + D_0^2\right),$$

where $M_{-1} := \mathbf{0}$. Therefore, we have

$$\sum_{t=0}^{T-1} \mathbb{E}\left[\|M_t\|_{\mathrm{F}}^2\right] \leq \left(\frac{\sigma^2}{b} + D_0^2\right) T.$$

This completes the proof. □

**Lemma C.2.** *Suppose Assumptions 2.1 and 2.2 hold and $\eta \leq \frac{1}{\lambda}$. Then, for all $t \in \mathbb{N}$,*

$$\sum_{t=0}^{T-1} \|W_t\|_{\mathrm{F}}^2 \leq \frac{\|W_0\|_{\mathrm{F}}^2}{\eta\lambda} + \frac{nT}{\lambda^2}.$$

*Proof.* From the definition of $W_t$, we have

$$\|W_t\|_{\mathrm{F}}^2 = \left\|(1-\eta\lambda)W_{t-1} - \eta\lambda \cdot \frac{1}{\lambda}O_t\right\|_{\mathrm{F}}^2$$
$$\leq (1-\eta\lambda)\|W_{t-1}\|_{\mathrm{F}}^2 + \frac{\eta n}{\lambda}$$
$$\leq (1-\eta\lambda)^t\|W_0\|_{\mathrm{F}}^2 + \frac{\eta n}{\lambda}\sum_{k=0}^{t}(1-\eta\lambda)^k$$
$$\leq (1-\eta\lambda)^t\|W_0\|_{\mathrm{F}}^2 + \frac{n}{\lambda^2}.$$

Hence,

$$\sum_{t=0}^{T-1} \|W_t\|_{\mathrm{F}}^2 \leq \frac{\|W_0\|_{\mathrm{F}}^2}{\eta\lambda} + \frac{nT}{\lambda^2}.$$

This completes the proof. □

**Theorem C.1** (Auxiliary Theorem for Muon with weight decay). *Suppose Assumptions 2.1 and 2.2 hold, and that Muon is run with $\eta \leq \frac{1}{\lambda}$. Then for all $t \in \mathbb{N}$,*

$$\sum_{t=0}^{T-1} \mathbb{E}\left[\|\nabla f(W_t)\|_{\mathrm{F}}\right] \leq \frac{f(W_0) - f(W_T)}{\eta} + \frac{1}{2}\sum_{t=0}^{T-1} \mathbb{E}\left[\|\nabla f(W_t) - C_t\|_{\mathrm{F}}^2\right] + \sqrt{r}\sum_{t=0}^{T-1}\mathbb{E}\left[\|\nabla f(W_t) - C_t\|_{\mathrm{F}}\right]$$

$$+ \frac{\lambda}{2}\sum_{t=0}^{T-1}\mathbb{E}\left[\|C_t\|_{\mathrm{F}}^2\right] + \lambda\left\{\frac{1 + 2(1+L\eta)\lambda}{2}\right\}\sum_{t=0}^{T-1}\mathbb{E}\left[\|W_t\|_{\mathrm{F}}^2\right] + (1+L\eta)nT,$$

*where* $\mathrm{rank}(C_t - \nabla f(W_t)) =: r_t \leq \max_{0 \leq t \leq T-1} r_t =: r.$

*Proof.* According to Assumption 2.1,

$$f(W_{t+1}) \leq f(W_t) + \langle \nabla f(W_t), W_{t+1} - W_t\rangle_{\mathrm{F}} + \frac{L}{2}\|W_{t+1} - W_t\|_{\mathrm{F}}^2$$

$$= f(W_t) - \eta\langle\nabla f(W_t), O_t + \lambda W_t\rangle_{\mathrm{F}} + \frac{L\eta^2}{2}\|O_t + \lambda W_t\|_{\mathrm{F}}^2$$

$$= f(W_t) - \eta\langle C_t, O_t + \lambda W_t\rangle_{\mathrm{F}} - \eta\langle\nabla f(W_t) - C_t, O_t + \lambda W_t\rangle_{\mathrm{F}} + \frac{L\eta^2}{2}\|O_t + \lambda W_t\|_{\mathrm{F}}^2$$

$$= f(W_t) - \eta\langle C_t, O_t\rangle_{\mathrm{F}} - \eta\lambda\langle C_t, W_t\rangle_{\mathrm{F}} - \eta\langle\nabla f(W_t) - C_t, O_t + \lambda W_t\rangle_{\mathrm{F}} + \frac{L\eta^2}{2}\|O_t + \lambda W_t\|_{\mathrm{F}}^2.$$

From Eq.equation 6, we have

$$-\eta\langle C_t, O_t\rangle_{\mathrm{F}} \leq \eta\sqrt{r}\|C_t - \nabla f(W_t)\|_{\mathrm{F}} - \eta\|\nabla f(W_t)\|_{\mathrm{F}}.$$

Applying the inequality $\|O_t + \lambda W_t\|_{\mathrm{F}}^2 \leq 2\|O_t\|_{\mathrm{F}}^2 + 2\lambda^2\|W_t\|_{\mathrm{F}}^2$, we obtain

$$f(W_{t+1}) \leq f(W_t) + \eta\sqrt{r}\|C_t - \nabla f(W_t)\|_{\mathrm{F}} - \eta\|\nabla f(W_t)\|_{\mathrm{F}} + \frac{\eta\lambda}{2}\|C_t\|_{\mathrm{F}}^2$$

$$+ \frac{\eta\lambda}{2}\|W_t\|_{\mathrm{F}}^2 + \frac{\eta}{2}\|\nabla f(W_t) - C_t\|_{\mathrm{F}}^2 + \frac{1+L\eta}{2}\eta\|O_t + \lambda W_t\|_{\mathrm{F}}^2$$

$$\leq f(W_t) + \eta\sqrt{r}\|C_t - \nabla f(W_t)\|_{\mathrm{F}} - \eta\|\nabla f(W_t)\|_{\mathrm{F}} + \frac{\eta\lambda}{2}\|C_t\|_{\mathrm{F}}^2$$

$$+ \frac{\eta\lambda}{2}\|W_t\|_{\mathrm{F}}^2 + \frac{\eta}{2}\|\nabla f(W_t) - C_t\|_{\mathrm{F}}^2 + (1+L\eta)\eta\left(\|O_t\|_{\mathrm{F}}^2 + \lambda^2\|W_t\|_{\mathrm{F}}^2\right)$$

$$\leq f(W_t) + \eta\sqrt{r}\|C_t - \nabla f(W_t)\|_{\mathrm{F}} - \eta\|\nabla f(W_t)\|_{\mathrm{F}} + \frac{\eta\lambda}{2}\|C_t\|_{\mathrm{F}}^2$$

$$+ \eta\lambda\left\{\frac{1 + 2(1+L\eta)\lambda}{2}\right\}\|W_t\|_{\mathrm{F}}^2 + \frac{\eta}{2}\|\nabla f(W_t) - C_t\|_{\mathrm{F}}^2 + (1+L\eta)\eta n.$$

By rearranging the terms and taking the expectation, we obtain

$$\mathbb{E}\left[\|\nabla f(W_t)\|_{\mathrm{F}}\right] \leq \frac{f(W_t) - f(W_{t+1})}{\eta} + \frac{1}{2}\mathbb{E}\left[\|\nabla f(W_t) - C_t\|_{\mathrm{F}}^2\right] + \sqrt{r}\mathbb{E}\left[\|\nabla f(W_t) - C_t\|_{\mathrm{F}}\right]$$

$$+ \frac{\lambda}{2}\mathbb{E}\left[\|C_t\|_{\mathrm{F}}^2\right] + \lambda\left\{\frac{1 + 2(1+L\eta)\lambda}{2}\right\}\mathbb{E}\left[\|W_t\|_{\mathrm{F}}^2\right] + (1+L\eta)n.$$

This completes the proof. □

## C.3 Proof of Theorem 3.2(i)

*Proof.* From $C_t := M_t$ and Lemmas A.1, A.2, C.1, C.2 and Theorem C.1, we find that

$$
\begin{aligned}
\sum_{t=0}^{T-1} \mathbb{E}\left[\|\nabla f(W_t)\|_{\mathrm{F}}\right] &\leq \frac{f(W_t) - f(W_T)}{\eta} + \frac{1}{2}\sum_{t=0}^{T-1}\mathbb{E}\left[\|\nabla f(W_t) - M_t\|_{\mathrm{F}}^2\right] + \sqrt{r}\sum_{t=0}^{T-1}\mathbb{E}\left[\|\nabla f(W_t) - M_t\|_{\mathrm{F}}\right] \\
&\quad + \frac{\lambda}{2}\sum_{t=0}^{T-1}\mathbb{E}\left[\|M_t\|_{\mathrm{F}}^2\right] + \lambda\left\{\frac{1+2(1+L\eta)\lambda}{2}\right\}\sum_{t=0}^{T-1}\mathbb{E}\left[\|W_t\|_{\mathrm{F}}^2\right] + (1+L\eta)nT \\
&\leq \frac{f(W_t) - f(W_T)}{\eta} + \frac{1}{2}\left\{\frac{2}{1-\beta}\|M_0 - \nabla f(W_0)\|_{\mathrm{F}}^2 + \frac{2(1-\beta)\sigma^2}{b}T + \frac{4L^2\eta^2 n}{(1-\beta)^2}T\right\} \\
&\quad + \sqrt{r}\left\{\frac{2\sqrt{2}}{1-\beta}\|M_0 - \nabla f(W_0)\|_{\mathrm{F}} + \sqrt{\frac{2(1-\beta)\sigma^2}{b}}T + \frac{2L\eta\sqrt{n}}{1-\beta}T\right\} \\
&\quad + \frac{\lambda}{2}\left(\frac{\sigma^2}{b} + D_0^2\right)T + \lambda\left\{\frac{1+2(1+L\eta)\lambda}{2}\right\}\left(\frac{\|W_0\|_{\mathrm{F}}^2}{\eta\lambda} + \frac{nT}{\lambda^2}\right) + (1+L\eta)nT.
\end{aligned}
$$

By taking the average over $t = 0, \dots, T-1$ and applying expectation, we obtain

$$
\begin{aligned}
\frac{1}{T}\sum_{t=0}^{T-1}\mathbb{E}\left[\|\nabla f(W_t)\|_{\mathrm{F}}\right] &\leq \frac{f(W_0) - f(W_T)}{\eta T} + \frac{\|M_0 - \nabla f(W_0)\|_{\mathrm{F}}^2}{(1-\beta)T} + \frac{2\sqrt{2r}\|M_0 - \nabla f(W_0)\|_{\mathrm{F}}}{(1-\beta)T} \\
&\quad + \left\{\frac{1+2(1+L\eta)\lambda}{2}\right\}\frac{\|W_0\|_{\mathrm{F}}^2}{\eta T} + \left(1 - \beta + \frac{\lambda}{2}\right)\frac{\sigma^2}{b} + \sqrt{\frac{2(1-\beta)r\sigma^2}{b}} \\
&\quad + \frac{2L^2\eta^2 n}{(1-\beta)^2} + \frac{2L\eta\sqrt{nr}}{1-\beta} + (1+L\eta)n + \left\{\frac{1+2(1+L\eta)\lambda}{2}\right\}\frac{n}{\lambda} + \frac{\lambda D_0^2}{2} \\
&= \mathcal{O}\left(\frac{1}{T} + \left(1 - \beta + \frac{\lambda}{2}\right)\frac{1}{b} + n\right).
\end{aligned}
$$

This completes the proof. $\qquad\square$

## C.4 Proof of Theorem 3.2(ii)

*Proof.* From the definition of $C_t$, we have Eq.equation 7, Eq.equation 8, and

$$
\begin{aligned}
\|C_t\|_{\mathrm{F}}^2 &= \|\beta M_t + (1-\beta)\nabla f_{\mathcal{S}_t}(W_t)\|_{\mathrm{F}}^2 \\
&\leq \beta\|M_t\|_{\mathrm{F}}^2 + (1-\beta)\|\nabla f_{\mathcal{S}_t}(W_t)\|_{\mathrm{F}}^2.
\end{aligned}
$$

Therefore, according to Theorem C.1, we find that

$$
\begin{aligned}
\sum_{t=0}^{T-1}\mathbb{E}\left[\|\nabla f(W_t)\|_{\mathrm{F}}\right] &\leq \frac{f(W_0) - f(W_T)}{\eta} + \frac{\beta}{2}\sum_{t=0}^{T-1}\mathbb{E}\left[\|\nabla f(W_t) - M_t\|_{\mathrm{F}}^2\right] + \frac{1-\beta}{2}\sum_{t=0}^{T-1}\mathbb{E}\left[\|\nabla f_{\mathcal{S}_t}(W_t) - \nabla f(W_t)\|_{\mathrm{F}}^2\right] \\
&\quad + \beta\sqrt{r}\sum_{t=0}^{T-1}\mathbb{E}\left[\|\nabla f(W_t) - M_t\|_{\mathrm{F}}\right] + (1-\beta)\sqrt{r}\sum_{t=0}^{T-1}\mathbb{E}\left[\|\nabla f_{\mathcal{S}_t}(W_t) - \nabla f(W_t)\|_{\mathrm{F}}\right] \\
&\quad + \frac{\lambda\beta}{2}\sum_{t=0}^{T-1}\mathbb{E}\left[\|M_t\|_{\mathrm{F}}^2\right] + \frac{\lambda(1-\beta)}{2}\sum_{t=0}^{T-1}\mathbb{E}\left[\|\nabla f_{\mathcal{S}_t}(W_t)\|_{\mathrm{F}}^2\right] \\
&\quad + \lambda\left\{\frac{1+2(1+L\eta)\lambda}{2}\right\}\sum_{t=0}^{T-1}\mathbb{E}\left[\|W_t\|_{\mathrm{F}}^2\right] + (1+L\eta)nT.
\end{aligned}
$$

Then, from Lemmas A.1, A.2, C.1, and C.2, we obtain

$$\sum_{t=0}^{T-1} \mathbb{E}\left[\|\nabla f(W_t)\|_{\mathrm{F}}\right] \leq \frac{f(W_0) - f(W_T)}{\eta} + \frac{\beta}{2}\sum_{t=0}^{T-1}\left\{\frac{2}{1-\beta}\|M_0 - \nabla f(W_0)\|_{\mathrm{F}}^2 + \frac{2(1-\beta)\sigma^2}{b}T + \frac{4L^2\eta^2 n}{(1-\beta)^2}T\right\}$$

$$+ \frac{(1-\beta)\sigma^2}{2b}T + \beta\sqrt{r}\left\{\frac{2\sqrt{2}}{1-\beta}\|M_0 - \nabla f(W_0)\|_{\mathrm{F}} + \sqrt{\frac{2(1-\beta)\sigma^2}{b}}T + \frac{2L\eta\sqrt{n}}{1-\beta}T\right\}$$

$$+ (1-\beta)\sqrt{r}\sqrt{\frac{\sigma^2}{b}}T + \frac{\lambda\beta}{2}\left(\frac{\sigma^2}{b} + D_0^2\right)T + \frac{\lambda(1-\beta)}{2}\left(\frac{\sigma^2}{b} + D_0^2\right)T$$

$$+ \lambda\left\{\frac{1 + 2(1+L\eta)\lambda}{2}\right\}\left(\frac{\|W_0\|_{\mathrm{F}}^2}{\eta\lambda} + \frac{nT}{\lambda^2}\right) + (1+L\eta)nT.$$

By taking expectations and averaging over the iterates, we obtain

$$\frac{1}{T}\sum_{t=0}^{T-1} \mathbb{E}\left[\|\nabla f(W_t)\|_{\mathrm{F}}\right] \leq \frac{f(W_0) - f(W_T)}{\eta T} + \frac{\beta\|M_0 - \nabla f(W_0)\|_{\mathrm{F}}^2}{(1-\beta)T} + \frac{2\beta\sqrt{2r}\|M_0 - \nabla f(W_0)\|_{\mathrm{F}}}{(1-\beta)T}$$

$$+ \left\{\frac{1 + 2(1+L\eta)\lambda}{2}\right\}\frac{\|W_0\|_{\mathrm{F}}^2}{\eta T} + \frac{(2\beta+1)(1-\beta) + \lambda}{2}\cdot\frac{\sigma^2}{b}$$

$$+ (\beta\sqrt{2(1-\beta)} + 1 - \beta)\sqrt{\frac{r\sigma^2}{b}}$$

$$+ \frac{2L^2\eta^2\beta n}{(1-\beta)^2} + \frac{2L\eta\beta\sqrt{nr}}{1-\beta} + (1+L\eta)n + \left\{\frac{1 + 2(1+L\eta)\lambda}{2}\right\}\frac{n}{\lambda} + \frac{\lambda D_0^2}{2}$$

$$= \mathcal{O}\left(\frac{1}{T} + \frac{(2\beta+1)(1-\beta) + \lambda}{2}\cdot\frac{1}{b} + n\right).$$

This completes the proof. $\qquad\square$

# D   Experimental Details

This section details the experimental setup for evaluating the optimizers. Our code will be available at `https://anonymous.4open.science/r/critical_batchsize_muon-F0BA/README.md`.

**Workloads and General Setup (Vision)**   All CIFAR experiments used ResNet-18 on CIFAR-10 and VGG-16 on CIFAR-100, trained from scratch. Training was performed for a fixed number of samples corresponding to 100 epochs at a batch size of 512. For any batch size $B$, the number of epochs was scaled as $E_B = 100 \times (512/B)$ to keep the number of seen samples constant. Each configuration was repeated with five random seeds, and we report mean and standard deviation of test accuracy.

**Workload and System Setup (Language Modeling: C4/ Llama3.1 (160M))**   For the language–modeling workload we trained Llama3.1 (160M) on the C4 dataset with sequence length 2048. We used the `torchtitan` codebase with PyTorch FSDP (full sharding) and activation checkpointing (`mode=full`). Gradient clipping with max norm 1.0 was applied. Due to sharding, Muon requires an additional *all-gather of gradients* before the optimizer step, which introduces one extra collective communication per step. We therefore report SFO-based metrics (steps × batch size) in addition to loss to separate algorithmic efficiency from system overhead.

Unless otherwise noted, the training budget was fixed to 3.2B tokens. At a base global batch size (GBS) of 128 this corresponds to 12,208 steps; for other GBS values the number of steps scales inversely to keep the token budget constant. We ran on up to 8 H100 GPUs with per-GPU micro-batch size 8 and no tensor, pipeline, or context parallelism (`TP=PP=CP=1`).

**Learning-Rate Schedules**   Vision workloads used a grid search at base batch size 512 and square-root scaling to other batch sizes for Muon and AdamW. Momentum SGD was tested with both square-root and linear scaling. For Llama3 160M, we used linear warmup followed by cosine decay. The base configuration uses 2000 warmup steps; in sweeps we also parameterized warmup as 10% of total steps to maintain a comparable schedule across different GBS.

**Hyperparameter Tuning Protocol (Vision)**   We performed an extensive grid search over base learning rates and weight decay. For $B \neq 512$, learning rates were scaled from the base value using $\eta_B = \eta_{512}\sqrt{B/512}$ for AdamW and Muon; for Momentum SGD we report both $\sqrt{B/512}$ and $B/512$ scaling. The shared search space is summarized in Table 4, and optimizer-specific spaces in Table 5.

Table 4: Shared hyperparameters for vision experiments.

| Hyperparameter | Value / Search Space |
|---|---|
| Model Architecture | ResNet-18, VGG-16 |
| Dataset | CIFAR-10, CIFAR-100 |
| Batch Size ($B$) | {2, 4, 8, 16, 32, 64, 128, 256, 512, 1024, 2048, 4096} |
| Base Epochs (at $B = 512$) | 100 |
| Random Seeds | 5 |

**Optimizer Configuration (Vision)**   For the hybrid Muon optimizer, we applied the Muon update to all convolutional layers in ResNet-18 (excluding the input stem). Biases, batch-norm parameters, and the final linear layer were updated by AdamW. The number of parameters updated by each component is listed in Table 6. For standard AdamW and Momentum SGD, the respective optimizer was applied to all parameters.

**Search Space (Language Modeling)**   For C4/ Llama3.1 (160M), we performed sweeps using grid search. GBS took values in {64, 256, 1024, 4096} when resources permitted. We tuned Muon and the AdamW baseline per GBS without automatic LR scaling.

Table 5: Optimizer-specific hyperparameter search spaces (vision). The base learning rate ($\eta_{512}$) is specified at a reference batch size of 512.

| Hyperparameter | AdamW | Momentum SGD | Muon |
|---|---|---|---|
| **Searched Parameters** | | | |
| Base Learning Rate ($\eta_{512}$) | {0.01, 0.001, 0.0001} | {0.001, 0.0005, 0.0001} | |
|   - Muon component | — | — | {0.01, 0.005, 0.001} |
|   - AdamW component | — | — | {0.01, 0.001, 0.0001} |
| Momentum | — | 0.9 | {0.7, 0.8, 0.9, 0.95, 0.99, 0.999} |
| Weight Decay ($\lambda$) | {0.1, 0.01, 0.001, 0.0001, 0} | {0.1, 0.01, 0.001, 0.0001, 0} | {0.1, 0.01, 0.001, 0.0001, 0} |
| **Fixed Parameters** | | | |
| Learning Rate Scaling | $\sqrt{B/512}$ | $\sqrt{B/512}$ (and $B/512$) | $\sqrt{B/512}$ |
| Adam $\beta_1, \beta_2$ | 0.9, 0.999 (default) | — | 0.9, 0.999 (default) |
| Adam $\epsilon$ | 1e-8 (default) | — | 1e-8 (default) |

Table 6: Number of parameters updated by each optimizer component in the Muon setup for ResNet-18, VGG-16 and Llama3.1 (160M).

| Model | Muon params | AdamW params |
|---|---|---|
| ResNet-18 | 11,157,504 | 16,458 |
| VGG-16 | 14,712,896 | 19,302,500 |
| Llama3.1 (160M) | 127,401,984 | 49,180,416 |

Table 7: Shared configuration for C4/ Llama3.1 (160M).

| Item | Value |
|---|---|
| Model | Llama3.1 (160M) (dim $= 768$, $n_{\text{layers}} = 18$, $n_{\text{heads}} = 12$) |
| Dataset | C4 |
| Sequence length | 2048 |
| Global batch size (GBS) | $\{32, 64, 128, 256, 512, 1024, 2048, 4096\}$ |
| Total tokens | $3.2 \times 10^9$ |
| Base steps (GBS=128) | 12,208 |
| Schedule | linear warmup then cosine decay (decay ratio 0.5) |
| Warmup | 2000 steps (base) or 10% of total steps (sweep) |
| Gradient clipping | max-norm $= 1.0$ |
| Parallelism | FSDP; TP=1, PP=1, CP=1 |
| Hardware | up to 8 H100 GPUs; local batch size $= 8$ |

Table 8: Optimizer search spaces for C4/ Llama3.1 (160M). Component-wise LRs are tuned per GBS.

| Hyperparameter | Muon | AdamW (baseline) |
|---|---|---|
| Muon LR | {0.005, 0.01, 0.02, 0.04} | — |
| Muon momentum $\beta$ | {0.8, 0.9, 0.95, 0.99} | — |
| Muon Nesterov | {True, False} | — |
| Muon weight decay | {0, 0.1} | — |
| AdamW LR (component or baseline) | {0.001, 0.002, 0.004, 0.008} | {0.001, 0.002, 0.004, 0.008} |
| AdamW weight decay | 0.1 | 0.1 |
| AdamW $\beta_1, \beta_2$ | 0.9, 0.999 | 0.9, 0.999 |
| AdamW $\epsilon$ | $1 \times 10^{-8}$ | $1 \times 10^{-8}$ |

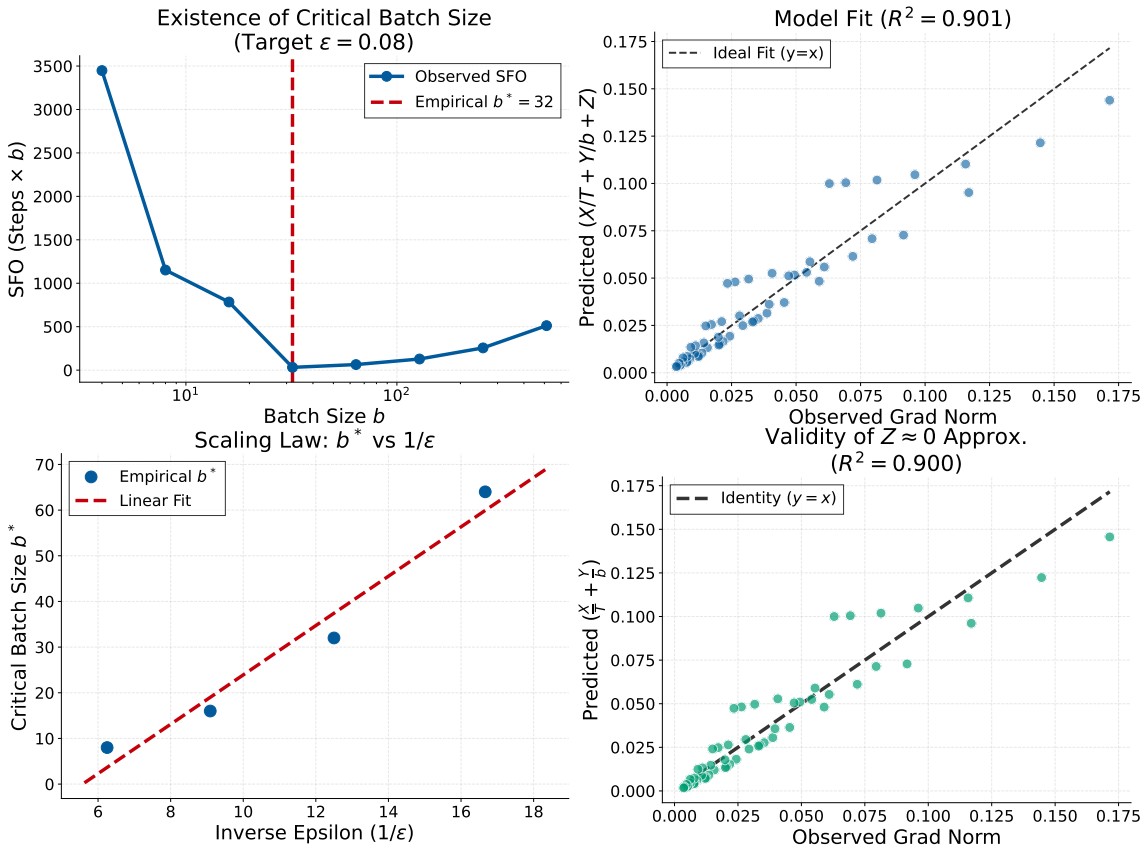

Figure 7: **Validation of critical batch size theory on a controlled full-Muon task. (Top Left)** SFO vs. batch size $b$ required to reach a target gradient norm $\varepsilon = 0.08$. A clear empirical optimum (critical batch size) is observed at $b^\star = 32$. **(Top Right)** Goodness-of-fit for the theoretical proxy $\bar{g}(T, b) \approx X/T + Y/b + Z$ against observed gradient norms ($R^2 \approx 0.901$). **(Bottom Left)** The empirical critical batch size $b^\star$ scales linearly with the inverse target precision $1/\varepsilon$, consistent with the theoretical prediction $b^\star \propto 1/\varepsilon$. **(Bottom Right)** Goodness-of-fit for the theoretical proxy $\bar{g}(T, b) \approx X/T + Y/b$ against observed gradient norms ($R^2 \approx 0.900$).

# E  Additional Diagnostics: Gradient-norm proxy and full-Muon toy setting

**Toy task and full-Muon configuration.**  To eliminate the gap between the theoretical analysis (which assumes Muon is applied to all parameters) and the practical hybrid optimizer used in vision experiments (Muon for Conv2D, AdamW for others), we conduct experiments on a controlled synthetic task. We employ a **Teacher-Student Tanh Regression** problem. A fixed teacher matrix $W^\star \in \mathbb{R}^{m \times n}$ and a student matrix $W \in \mathbb{R}^{m \times n}$ are initialized with entries drawn from $\mathcal{N}(0, 1/n)$. At each step $t$, we sample inputs $x_t \sim \mathcal{N}(0, I_n)$ and generate labels $y_t = \tanh(x_t(W^\star)^\top) + \xi_t$, where $\xi_t \sim \mathcal{N}(0, \sigma^2 I)$ represents label noise. The student minimizes the loss $\mathcal{L}(W) = \frac{1}{2}\|\tanh(x_t W^\top) - y_t\|^2$. We set the dimensions to $m = 256, n = 128$, and the noise level to $\sigma = 0.1$. Unlike the large-scale experiments, here Muon (Algorithm 1) is applied to the entire matrix $W$ with learning rate $\eta = 0.05$, momentum $\beta = 0.95$ (Nesterov), and 5 Newton-Schulz iterations.

**Measured quantities and stopping rule.**  To rigorously validate the theoretical convergence bound $\mathbb{E}[\|\nabla\mathcal{L}\|] \lesssim \frac{X}{T} + \frac{Y}{b} + Z$, we track the *cumulative mean* of the gradient norm, denoted as $\bar{g}_t = \frac{1}{t}\sum_{i=1}^{t}\|\nabla\mathcal{L}(W_i)\|_F$. We define the convergence step $T_\varepsilon(b)$ as the first step $t$ where $\bar{g}_t \leq \varepsilon$. The stochastic first-order oracle (SFO) complexity is computed as $\text{SFO} = b \cdot T_\varepsilon(b)$. We perform a grid search over batch sizes $b \in \{4, 8, \ldots, 512\}$.

**Results and Validity of Approximations.** We confirm three key findings in this controlled setting: (1) **Existence of Critical Batch Size:** As shown in Figure 7 (Top Left), the SFO complexity exhibits a clear convex shape with respect to $b$, identifying an empirical optimum $b^\star$. (2) **Model Fit:** The observed data fits the theoretical proxy $\frac{X}{T} + \frac{Y}{b} + Z$ with high accuracy ($R^2 > 0.90$), validating our convergence analysis. (3) **Justification for** $Z \approx 0$**:** To address the theoretical concern regarding the non-vanishing term $Z$, we compared the full fit against a simplified model $\frac{X}{T} + \frac{Y}{b}$ (forcing $Z = 0$). Figure 7 (Bottom Right) demonstrates that the simplified model predicts the observed gradient norms with virtually identical accuracy to the full model in the regime of interest. This empirically justifies our derivation of the critical batch size $b^\star \propto 1/\varepsilon$ by treating $Z$ as negligible during the primary optimization phase.

# F   Additional Results (Vision)

This section provides supplementary results to support the analysis in the main text.

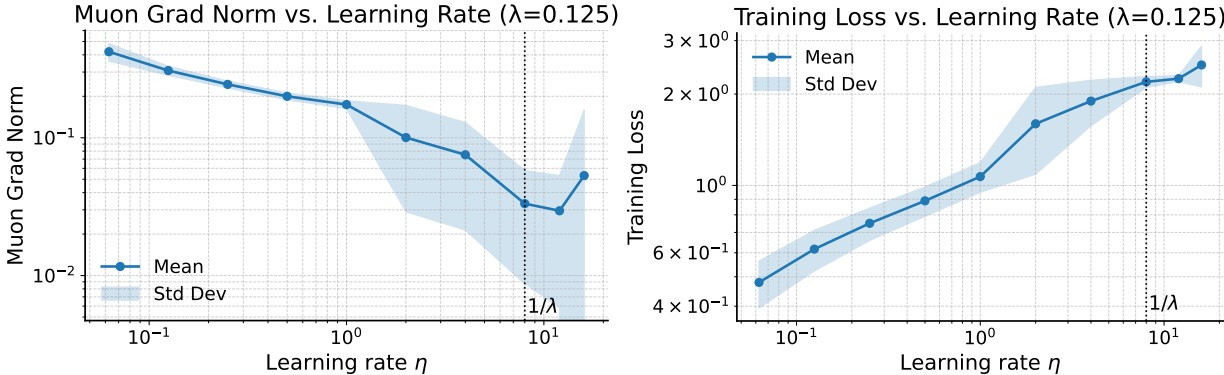

Figure 8:   Empirical validation of the stability condition derived in Proposition 3.2. The plots show the final gradient norm (left) and training loss (right) for ResNet-18 on CIFAR-10, trained with Muon using various learning rates ($\eta$) and a fixed weight decay $\lambda = 0.125$. The vertical dashed lines mark the theoretical stability threshold $\eta = 1/\lambda$. Training was most stable and achieved the best performance near this threshold, consistent with our theoretical analysis.

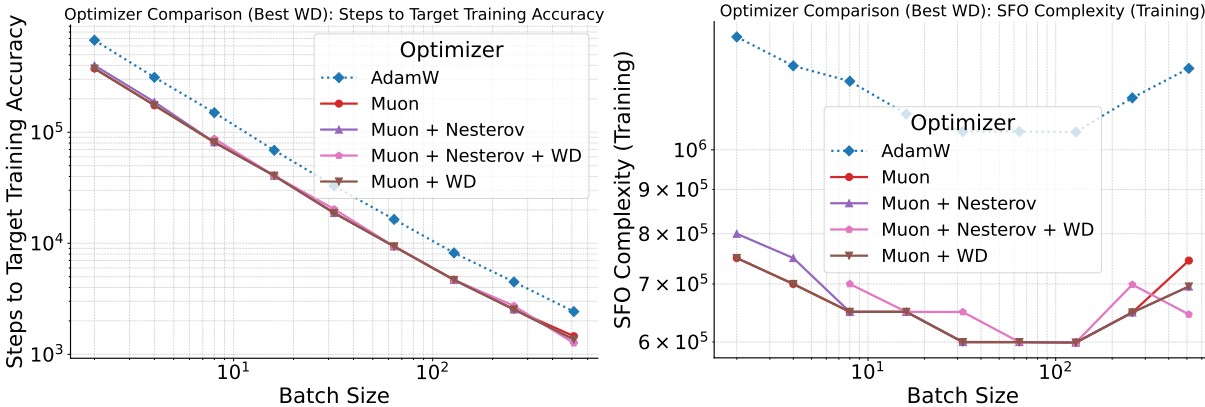

Figure 9:   Analysis of batch size scaling and SFO complexity for ResNet-18 on CIFAR-10. (Left) Number of steps required to reach target training accuracy (95%). (Right) SFO complexity to reach the same accuracy. Muon exhibited superior scaling to large batch sizes, and its critical batch size (which minimizes SFO complexity (training)) was smaller than that of AdamW.

**Convergence (Vision)**   We conducted experiments under various weight decay configurations to examine whether the upper bound of the learning rate is determined by the weight decay ($\lambda = 0.125$), similar to the analysis presented in Figure 1 of the main text ($\lambda = 0.0625$). As shown in Figure 8, the results are consistent with the earlier findings, confirming that the gradient norm begins to increase once the learning rate exceeds $1/\lambda$.

Figure 10 compares convergence rates for ResNet-18 on CIFAR-10 across three batch sizes. This complements Figure 2 by providing a more detailed view of the effect of batch size.

**Critical Batch Size (Vision)**   In the main text, we defined the critical batch size as the smallest batch size that reaches the test target accuracy in the fewest steps. For comparison, we also report results using the training target accuracy in Figure 9. While the overall trends remain similar, the gap between Adam and Muon becomes significantly larger in this setting.

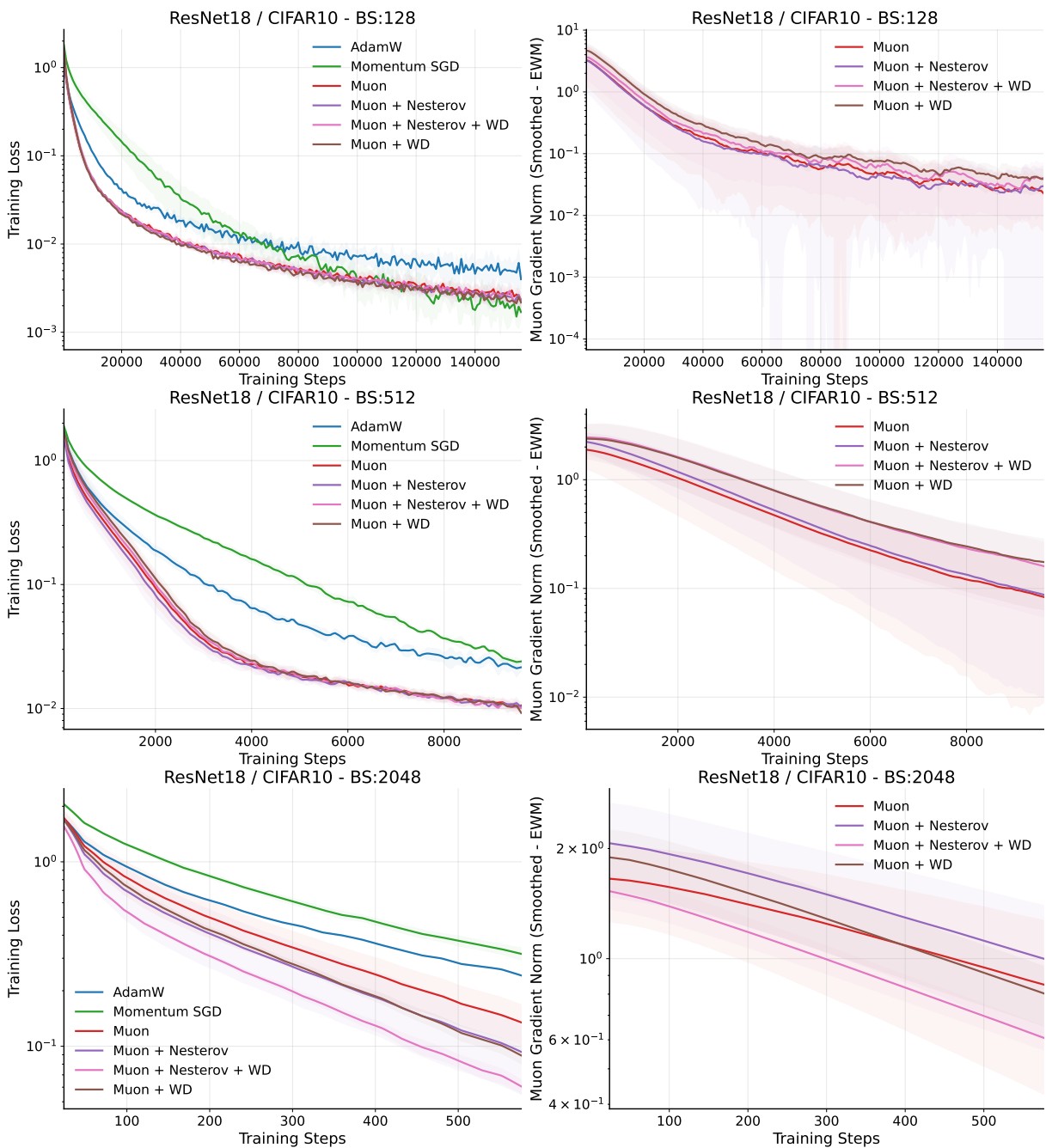

Figure 10: Convergence rate comparison for ResNet-18 on CIFAR-10 across three batch sizes (128, 512, and 2048 from top to bottom). Each row compares training loss (left) and smoothed gradient norm (right) for the Muon variants and baselines. These results supplement Figure 2 and confirm that the observed performance trends hold across a range of batch sizes.

In the main text, Figure 3 presents results exclusively for Adam and Muon. For a broader comparison, results for Momentum SGD are included and shown in Figure 11.

**Additional Vision Workload** Due to space limits, the main text focuses on ResNet-18/CIFAR-10. Figure 12 reports VGG-16/CIFAR-100, which shows the same qualitative behavior.

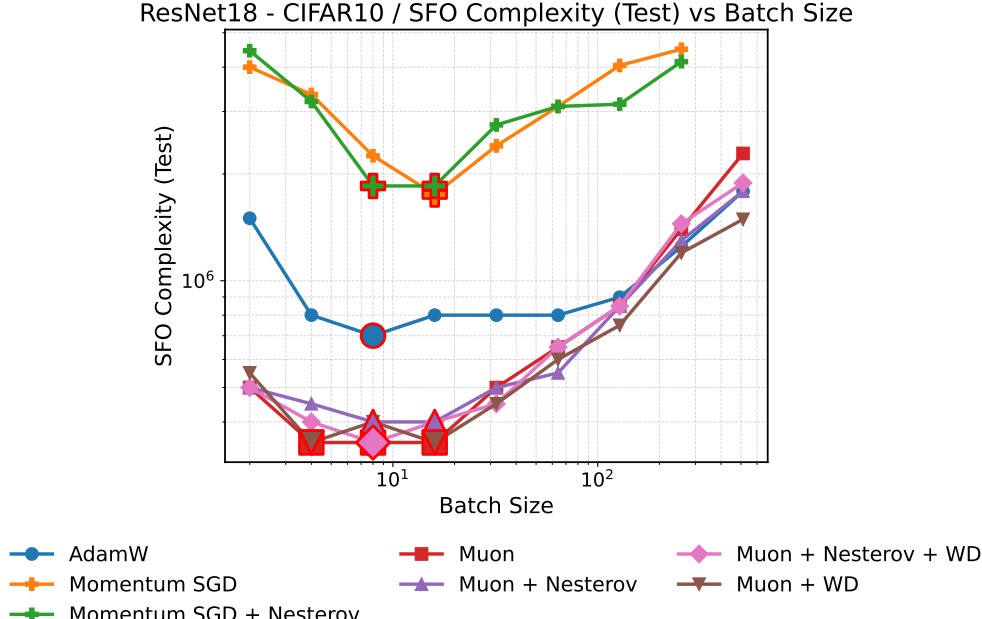

Figure 11: Optimizer comparison via analysis of batch size scaling and SFO complexity (test) for ResNet-18 on CIFAR-10.

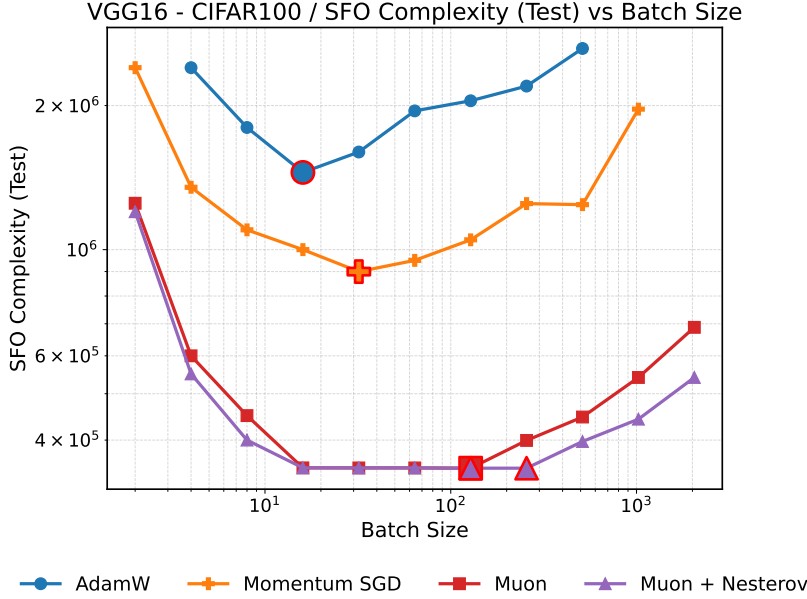

Figure 12: Optimizer comparison via analysis of batch size scaling and SFO complexity (test) for VGG-16 on CIFAR-100.

**Ablation Study**  Finally, we conducted an ablation study to examine the relationship between the weight decay parameter $\beta$ and the learning rate across four batch sizes. Figure 13 illustrates how Muon's weight decay and learning rate affect the loss for each batch size. For the training loss (top row), lower weight decay consistently corresponds to reduced loss. Similarly, the smallest learning rates within the explored range are preferred. For the test loss (bottom row), smaller learning rates yield better results. However, a clear inflection point emerges in weight decay at around $10^{-1}$, or approximately $10^{-2}$ for larger batch sizes, indicating that these weight decay settings minimize the test loss.

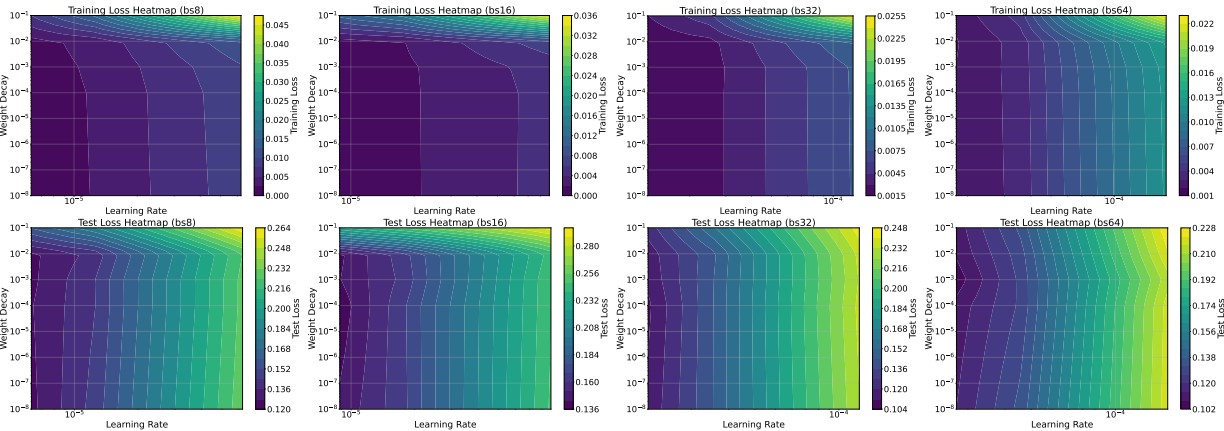

Figure 13: Ablation study on the impact of weight decay and learning rate on loss for Muon across different batch sizes (8, 16, 32, and 64, from left to right). Each column compares training loss (top) and test loss (bottom) across various configurations of weight decay and learning rate. The training loss consistently favors smaller weight decay values and the lowest learning rates within the explored range. For test loss, optimal performance is achieved at weight decay values around $10^{-1}$, shifting toward approximately $10^{-2}$ at larger batch sizes.

# G   Additional Results (Language Modeling)

We provide supplemental plots for the LLM experiments discussed in the main text. Figure 14 shows final loss and SFO complexity versus batch size. Muon is consistently better than AdamW and the gap widens at large batch sizes. Nesterov momentum and weight decay do not produce systematic gains in this setting.

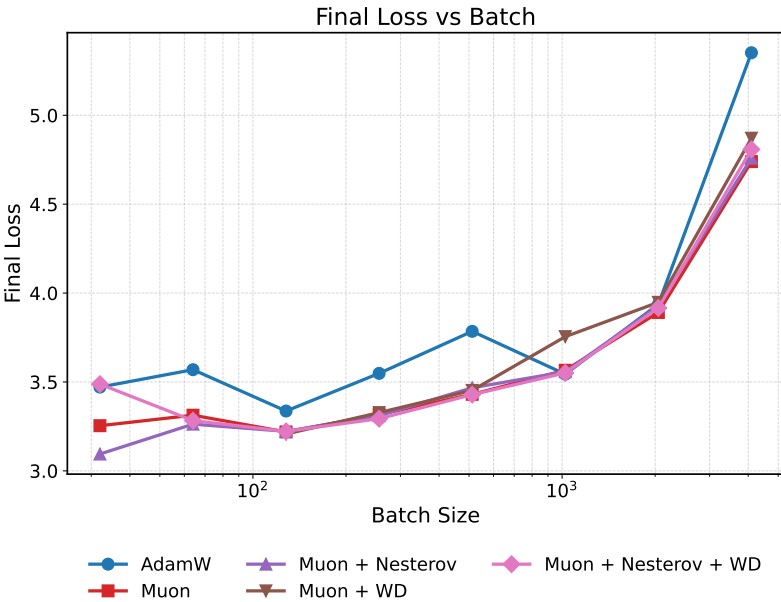

Figure 14: C4/ Llama3.1 (160M). Final training loss versus batch size. Muon outperforms AdamW across all batch sizes, with a larger margin at bigger batches.

To examine the role of momentum, we swept $\beta$ under two configurations: with and without Nesterov (weight decay fixed at zero unless indicated). Figure 15 shows that the best trade-off is near $\beta = 0.95$. As $\beta$ increases, the critical batch size decreases, but very small or very large values of $\beta$ degrade both loss and SFO.

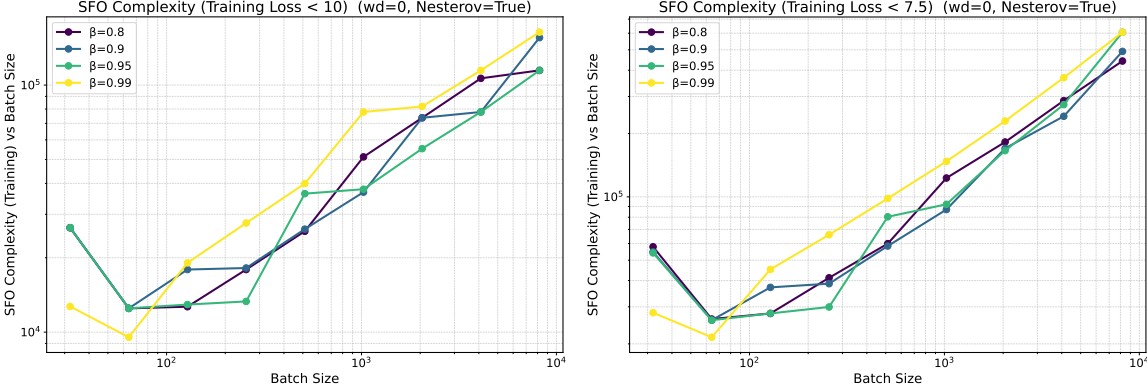

Figure 15: Effect of momentum $\beta$ on C4/ Llama3.1 (160M). SFO for target loss of 10.0 (left) and SFO for target loss of 7.5 (right) across batch sizes under (WD=0, Nesterov=True). The optimum is near $\beta = 0.95$; excessive or too small momentum harms both metrics.

**Training Curves for Muon on Llama3.1 (160M)**   To further illustrate Muon's optimization dynamics on large-language-model workloads, we report full training curves for Llama3.1 (160M) trained on C4 at multiple batch sizes. Figure 17 plots the training loss against the number of steps for Muon and AdamW. Across all batch

sizes, Muon exhibits faster loss reduction in early training and consistently reaches lower final loss within the same token budget. These results complement Figures 5 and 6, demonstrating that the advantages of Muon are visible not only in SFO complexity but also in the raw optimization trajectory.

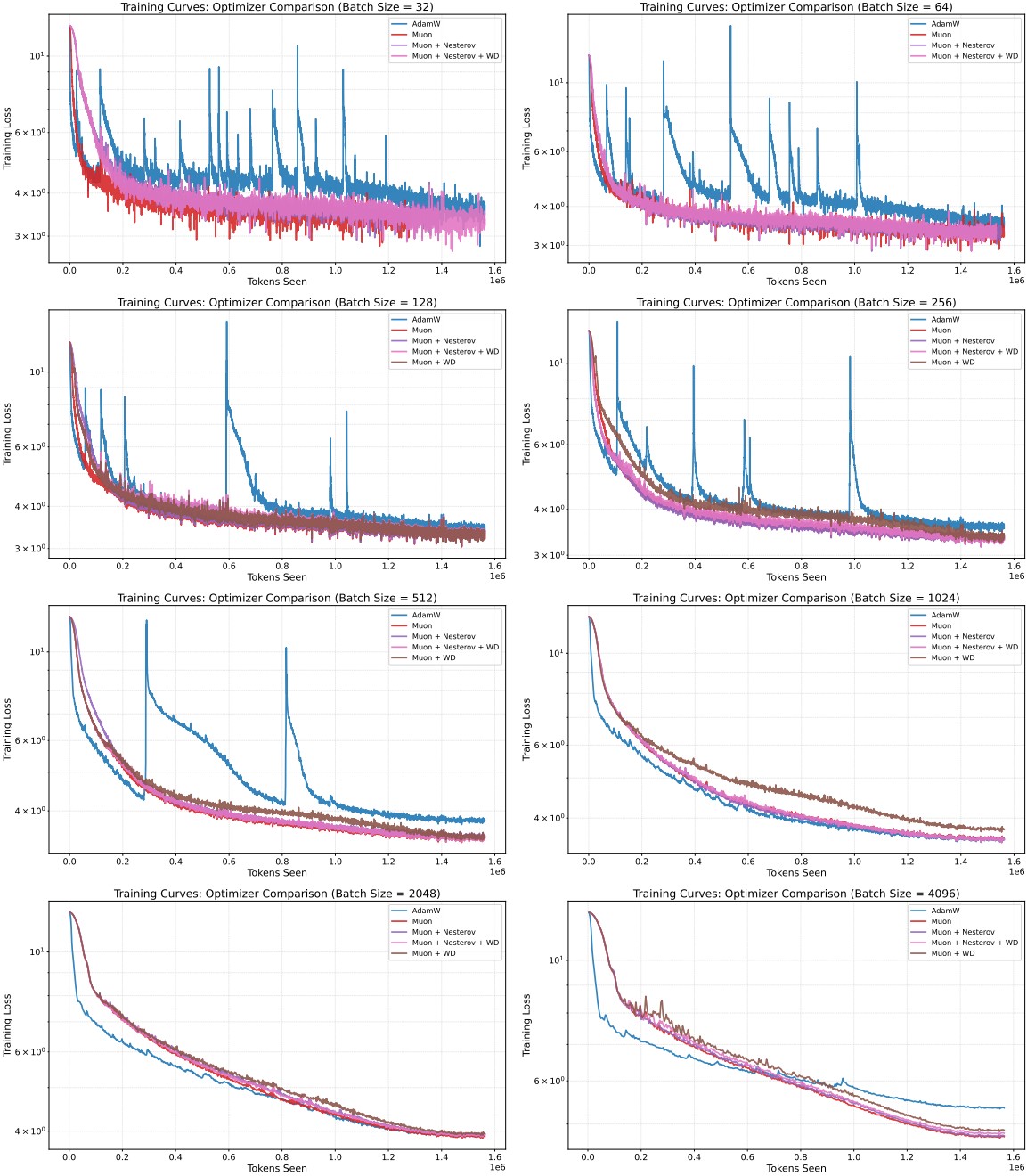

Figure 16: Training loss curves for Muon vs. AdamW on Llama3.1 (160M)/ C4. Each plot corresponds to a global batch size of 32, to 4096 (top to bottom). Muon variants consistently achieves faster loss reduction and better final loss across all batch sizes.

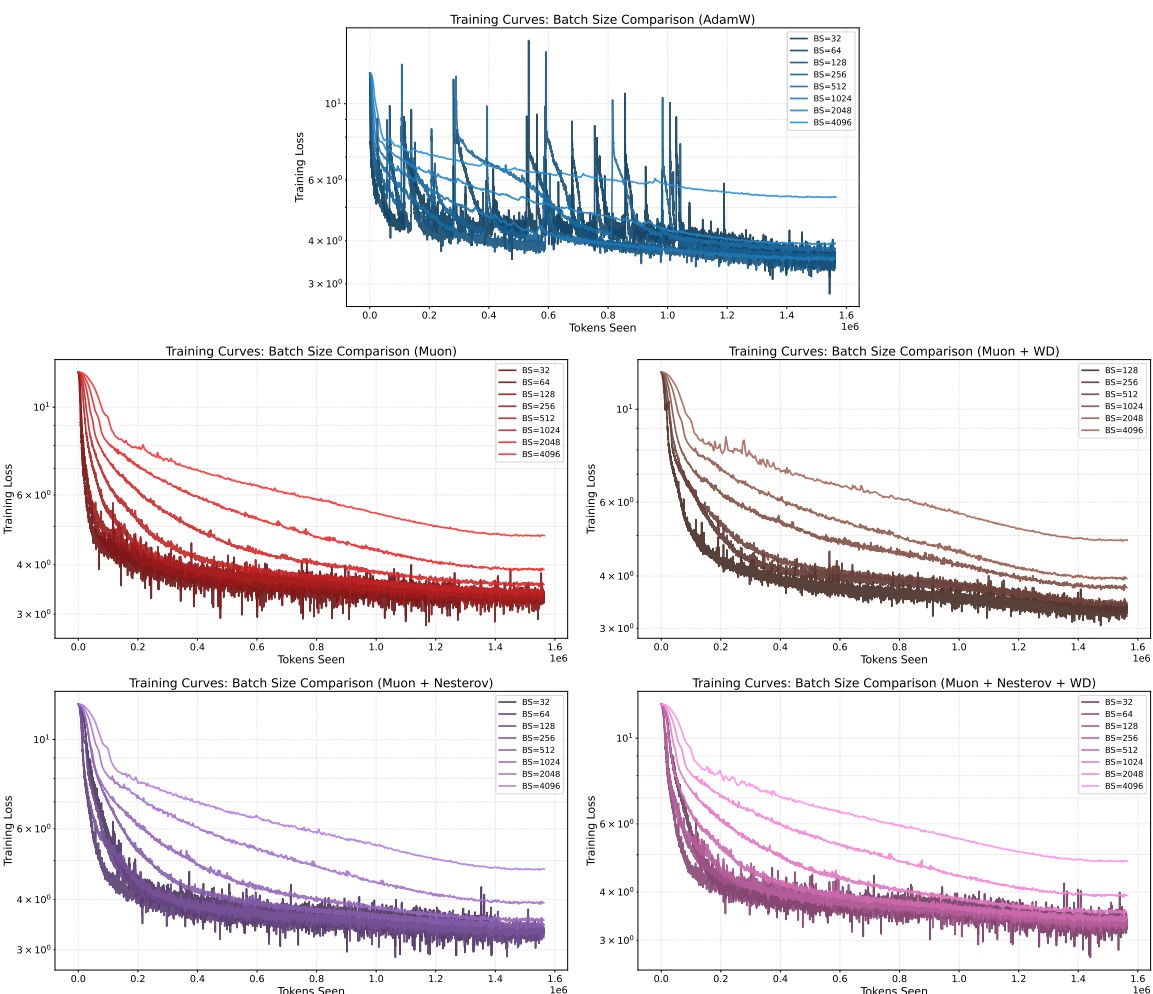

Figure 17: Training loss curves for batch size on Llama3.1 (160M)/ C4. Each plot corresponds to a Adam, Muon, Muon+WD, Muon+Nesterov, Muon+Nesterov+WD (top to bottom). Small batch size consistently achieves better loss better final loss across optimizers.

