# OpenReview forum: "Convergence Bound and Critical Batch Size of Muon Optimizer"
_TMLR — Rejected by TMLR_

### Review · Reviewer_d7rw · 2025-12-09

**Summary Of Contributions:**

This paper studies the convergence of Muon under the stochastic setting where the objective is nonconvex and smooth. The four variations with and without momentum and weight decay are all studied in this paper. Moreover, based on the derived upper bound, the authors analyzed the critical batch size, that is, the batch size that can minimize the total sample complexity for achieving a certain accuracy epsilon.

Strength: 1. The convergence rates in all the settings are presented. The most practical case with both weight decay and momentum is studied.
2. Extensive experiments are performed.

Weakness:
1. There are lots of recent works studying the convergence of Muon, and I have seen papers with similar convergence guarantees. Given existing works, the results in this work look a bit incremental. Is any rate in this work the "first" rate developed?

2. The notion of the critical batch size is not well-motivated. Compared to some other rates I am aware of, the upper bounds in this work all have an O(n) term, which I believe is because the assumption is on the Frobenius norm. However, this term shouldn't be dropped since it is clearly the dominating term. Given this, the derivation in Section 4.1 does not make much sense to me. Also, an upper bound can be loose, so it may not reflect what the real expected gradient norm is. Equations (4) and (5) are not convincing. Additionally, the experiments only validate that the critical batch size decreases as beta tends to 1, but not the formula derived. Given existing works on the convergence guarantees of Muon, the critical batch size is the main contribution of this work. I view this as the biggest weakness.

**Audience:**

Yes

**Audience Explanation:**

People interested in understanding the recent popular optimizer Muon may find this paper useful.

**Broader Impact Concerns:**

No extra concerns.

**Claims And Evidence:**

Yes

**Claims Explanation:**

Proofs of all the statements are provided. The authors also provide numerical experiments for training deep neural networks and language models, which help support the theoretical results.

**Requested Changes:**

1. Page 2, line 2. There are two Bernsteins.
2. Equation (3), the "<T" part in the definition of T(b) is confusing.

---

> ### Author Response · Authors · 2026-01-25
>
> We deeply appreciate your thorough reading and high regard for our paper. In response to your valuable comments, we have prepared the following reply. We are ready to respond to any comments. Please feel free to leave a comment.
>
> ---
>
> **Weakness 1:** There are lots of recent works studying the convergence of Muon, and I have seen papers with similar convergence guarantees. Given existing works, the results in this work look a bit incremental. Is any rate in this work the "first" rate developed?
>
> **Reply to Weakness 1:**
> We are the first to evaluate the gradient norm of Muon incorporating Nesterov momentum and weight decay.
>
> **Where in tha paper:** Table 3 has been added in Section 6.
>
> ---
>
> **Weakness 2-1:** The notion of the critical batch size is not well-motivated. Compared to some other rates I am aware of, the upper bounds in this work all have an O(n) term, which I believe is because the assumption is on the Frobenius norm. However, this term shouldn't be dropped since it is clearly the dominating term. Given this, the derivation in Section 4.1 does not make much sense to me. Also, an upper bound can be loose, so it may not reflect what the real expected gradient norm is. Equations (4) and (5) are not convincing.
>
> **Reply to Weakness 2-1:**
> Let us clarify that we are not ignoring or omitting $Z$.
> The reviewer's comment is that we should consider $X/T + Y/b + Z < \epsilon_{\text{tot}}$. According to prior research [1], under this condition, we can obtain $b^\star = \frac{2Y}{\epsilon_{\text{tot}} - Z} > \frac{2Y}{\epsilon_{\text{tot}}}$.
> However, since $Y/b$ and $Z$ do not depend on $T$, the upper bound of the gradient norm converges to $Y/b + Z$ as $T \to \infty$.
> Therefore, when training is sufficiently complete, we can assume $X/T + Y/b < \epsilon$ holds. Note that $\epsilon_{\text{tot}}\neq\epsilon$, and $\epsilon := \epsilon_{\text{tot}} - Z$. Under this equation, as stated in the main body, we can obtain $b^\star = \frac{2Y}{\epsilon}$. Compared to the above equation, the relationship between the hyperparameter and critical batch size $b^\star$ is clearer, and the theory does not break down merely due to a different threshold choice.
> The confusion arose because $\epsilon$ appeared to be a threshold for the average gradient norm. We refined the notation and added a remark to address this issue.
>
> **Where in tha paper:** Remark 4.1 has been added in Section 4.
>
> ---
>
> **Weakness 2-2:** Additionally, the experiments only validate that the critical batch size decreases as beta tends to 1, but not the formula derived.
>
> **Reply to Weakness 2-2:**  We agree that validating only qualitative trends (e.g., “$b^\star$ decreases as $\beta \to 1$”) is insufficient. To provide a quantitative check, we now directly test the proxy form that underlies Eq. (4)(5):
>
> - In a controlled full-Muon setting, we fit the observed average gradient norms to
>   $\bar g(T,b)\approx X/T + Y/b + Z$
>   using a linear regression in $(1/T,1/b,1)$, and report goodness-of-fit (e.g., $R^2=0.901$).
> - From the fitted coefficients $\hat Y,\hat Z$, we compute a **predicted** critical batch size
>   $b^\star_{\text{pred}}=\frac{2\hat Y}{\varepsilon_{\text{tot}}-\hat Z},$
>   and compare it to the empirically SFO-minimizing batch size.
> - We additionally show that, above the floor ($\bar \epsilon_{\text{tot}}>\hat Z$), the scaling
>   $b^\star \propto 1/(\bar \epsilon_{\text{tot}} - \hat Z)$ is observed, and that the relationship breaks down as $\bar \epsilon_{\text{tot}}$ approaches the floor, consistent with the model.
>
> **Where in the paper:** Added as a new paragraph in Section 5 under “Critical Batch Size”: “Quantitative validation of the $X/T+Y/b+Z$ proxy and predicted $b^\star$”, with full details/plots in the appendix diagnostics section (Figure 7).
>
> ---
>
> **Reference**
>
> [1] N. Sato and H. Iiduka, Existence and Estimation of Critical Batch Size for Training Generative Adversarial Networks with Two Time-Scale Update Rule, ICML2023.

---

> > ### Comment · Reviewer_d7rw · 2026-02-12
> >
> > I submitted my recommendation a while ago, but I forgot to reply here. Thank you for your response.  Item 1 is important, and this contribution should be made more pronounced. Meanwhile, I was not satisfied with the explanation of weakness 2-1. It is apparent to me that Z dominates eps, and eps_tol is mainly Z, which is O(n), so I care much less about eps than eps_tol.

---

> > > ### Author Response · Authors · 2026-02-16
> > >
> > > Thank you for your reply. As you mentioned, $\epsilon_\text{tot}$ is primarily $Z$.
> > >
> > > By using $\epsilon = \epsilon_\text{tot} - Z$ as the threshold, we aim to clarify the relationship between the critical batch size and the hyperparameters.
> > > Both approaches are essentially the same, differing only in the threshold value. That is, $X/T + Y/b < \epsilon$ and $X/T + Y/b + Z < \epsilon_{\text{tot}}$ are almost the same.
> > >
> > > Thank you for your confirmation.

---

### Review · Reviewer_mpr6 · 2025-12-29

**Summary Of Contributions:**

The paper studies the Muon optimizer, which forms an update direction by orthogonalizing a momentum matrix (via the polar factor of the momentum matrix), and analyzes four variants: with/without Nesterov momentum and with/without (decoupled) weight decay (Algorithm 1; Theorems 3.1–3.2; Table 1).

**Main contributions claimed/provided:**
- Convergence-style upper bounds on the average expected full gradient norm, summarized in Table 1, for the four Muon variants (Theorems 3.1 and 3.2).
- With weight decay, a stability/stepsize condition η ≤ 1/λ is assumed and used to prove boundedness of parameter norms and (via L-smoothness) boundedness of gradient norms (Propositions 3.1–3.2), and the paper empirically explores learning-rate sweeps around η = 1/λ (Figure 1).
- A derivation of a “critical batch size” for Muon (Section 4; Propositions 4.1–4.3), using an SFO-complexity proxy (steps × batch size), with formulas depending on β, λ, σ², and ε.
- Experiments on vision (ResNet-18/CIFAR-10; VGG-16/CIFAR-100) and language modeling (Llama3.1 160M/C4) reporting steps-to-target and SFO vs batch size, and momentum sweeps (Figures 2–6 and appendices).

**Key strengths:**
- Timely topic; Muon is widely discussed and understanding its hyperparameter interactions (β, λ, η, batch size) is useful.
- The paper includes a practical variant (Nesterov + weight decay) and provides multi-workload empirical plots (vision + LM) including batch-size sweeps.

**Most important weaknesses:**
- The main bounds include an additive dimension-dependent “+ n” term (Table 1; Theorem 3.1/3.2), so the narrative “rate ≈ 1/T” is not supported as stated.
- The “critical batch size” derivation explicitly drops a non-vanishing term (Section 4.1: discarding Z), making the resulting optimizer-specific conclusions unclear.

**Additional Comments:**

- The empirical plots (batch-size sweeps, momentum sweeps) are useful. However, given the theory’s use of a stationarity proxy (average gradient norm bound), it would strengthen the paper to report at least one evaluation that is closer to that proxy (even if approximate/smoothed).
- The claim that η ≤ 1/λ is “necessary” should be softened unless necessity is proved; currently it is an assumption required for Propositions 3.1–3.2 and Theorem 3.2, and empirically it appears to correlate with stability in Figure 1.

**Audience:**

Yes

**Audience Explanation:**

Muon (and more broadly, gradient orthogonalization / norm-constrained update rules) is actively used and discussed, and practitioners care about stability and scaling behavior with respect to η, β, λ, and batch size. The paper provides (i) a unified treatment of four practical Muon configurations and (ii) multi-workload empirical batch-size sweeps and momentum sweeps.
The theory needs tightening, these topics and experiments are of clear interest to parts of the TMLR audience working on optimization and large-scale training.

**Broader Impact Concerns:**

No ethical implications.

**Claims And Evidence:**

Yes

**Claims Explanation:**

1) The stated “≈ 1/T” convergence interpretation is not supported by the theorems as written.
- Table 1 and Theorems 3.1–3.2 include an additive “+ n” term in the upper bound (i.e., a term that does not decay with T). Consequently, for fixed η, the bound does not imply the average gradient norm can be made arbitrarily small by increasing T.
- However, immediately after Theorem 3.1 the text states: “These results establish that Muon reduces the norm of the full gradient at a rate of approximately 1/T …”. This statement is not implied by a bound containing a non-vanishing +n floor.

2) The analysis does not yet cover the implemented algorithm because orthogonalization is assumed exact.
- Algorithm 1 computes Ot via NewtonSchulz5(Ct), and the paper explicitly states: “Our theoretical analysis assumes that Ot := NewtonShulz5(Ct) satisfies Eq. (1).”
- Since Eq. (1) defines the exact polar-factor solution, the proofs analyze an idealized “oracle Muon” unless the approximation error of the truncated Newton–Schulz iteration is bounded and propagated.

3) The “critical batch size” theory is derived after discarding a term that the earlier theory indicates is non-vanishing.
- Section 4.1 starts from a bound of the form X/T + Y/b + Z and then states: “we exclude the term Z” to derive T(b) and optimize SFO complexity. This makes the final b* = 2Y/ε (and the subsequent formulas in Table 2) a consequence of the simplified proxy rather than the original bound.
- Since Z is precisely the term responsible for the bound’s nonzero floor (including the dimension-dependent contribution), it is unclear that the resulting “critical batch size” is meaningfully justified by the paper’s convergence bound.

Overall: the empirical results are suggestive, but the theoretical claims (as stated) are not yet supported by accurate/convincing evidence, mainly due to (i) the non-vanishing +n term and its interpretation, (ii) the idealization of Newton–Schulz orthogonalization, and (iii) discarding Z in the critical-batch analysis.

**Requested Changes:**

[CRITICAL] Fix the “convergence rate ≈ 1/T” narrative to match what is actually proved.
- At minimum, explicitly acknowledge that the main bounds contain a non-vanishing +n term (Table 1; Theorems 3.1–3.2) and therefore do not imply convergence to stationarity for constant η.
- Either (A) reframe the guarantee as convergence to an (η,β,λ,n)-dependent neighborhood, or (B) provide a stepsize schedule (e.g., diminishing ηt) under which the non-vanishing term is removed/controlled and the bound supports a true stationarity statement.

[CRITICAL] Address the source of the +n term and tighten the dimension dependence if possible.
- The +n term arises from bounding ||Ot||F^2 = n and from dual-norm manipulations in the auxiliary theorems (Appendix B/C). Please audit whether any steps are unnecessarily loose (e.g., nuclear/Frobenius inequalities and rank factors), and state the sharpest dependence you can support.
- If the +n dependence is unavoidable under the current proof strategy, state this clearly and adjust the claims accordingly.

[CRITICAL] Either incorporate Newton–Schulz truncation error in the theory, or clearly scope the results to an idealized algorithm.
- The paper currently assumes Ot := NewtonShulz5(Ct) exactly satisfies Eq. (1). To connect to the implemented optimizer, provide a bound on ||NewtonSchulz5(Ct) − Polar(Ct)|| (under stated conditions) and propagate it through the descent argument.
- If this is out of scope, then explicitly separate “ideal Muon” (exact polar step) from the practical implementation and limit theoretical conclusions accordingly.

[CRITICAL] Rework the critical batch size derivation so it does not rely on discarding Z (Section 4.1).
- Either keep Z in the stopping condition and show what b* becomes (or when the simplified b* ≈ 2Y/ε is valid), or justify with explicit assumptions why Z is negligible for the regime of interest.
- Align the experimental validation with the theoretical stopping proxy: currently experiments use accuracy/loss targets, whereas theory uses an upper bound on an average gradient norm. At least one additional diagnostic (e.g., steps to reach a fixed gradient-norm threshold, or a clearer argument linking the chosen loss/accuracy targets to ε) would improve evidence.

[STRENGTHENING] Clarify novelty and positioning relative to existing Muon convergence analyses cited in Section 6.
- Section 6 cites multiple convergence analyses/frameworks (e.g., Pethick et al. (2025), Shen et al. (2025), Kovalev (2025), Li & Hong (2025)). Add a short comparison table listing assumptions (exact/approx polar, stationarity measure, dimension dependence, momentum/weight-decay coverage) and explicitly state what this paper adds (Nesterov + decoupled weight decay) beyond prior work.

[STRENGTHENING] Clarify the mismatch between the theoretical setting and the experimental optimizer.
- Appendix D states a hybrid optimizer is used in vision experiments (Muon applied to certain parameter blocks, AdamW to others). Please discuss whether the theorems are intended to model this hybrid scheme, or explicitly list it as a limitation.

Minor presentation/consistency fixes.
- “NewtonShulz” vs “NewtonSchulz” naming consistency; ensure proof notation is consistent (there appear to be occasional typographical inconsistencies in appendices).
- The paper’s definition/usage of “critical batch size” should be consistent across the main text and appendix (Section 4 defines it via SFO minimization, but I'm a little puzzled by what you mean later).

---

> ### Author Response · Authors · 2026-01-25
>
> We sincerely appreciate for your review of our manuscript. We greatly appreciate your insights, and we have carefully incorporated your suggestions into our revised manuscript. Below, we present our point-by-point responses to your comments.
>
> ---
>
> **Requested Change 1:** Fix the "convergence rate $\approx 1/T$" narrative to match what is actually proved.
>
> **Reply:** As you pointed out, our bounds include terms independent of $T$, so they do not guarantee convergence to a stationary point. We fully acknowledge your observation and have revised the manuscript accordingly (see the paragraph below Theorem 3.1).
>
> ---
>
> **Requested Change 2:** Address the source of the $+n$ term and tighten the dimension dependence if possible.
>
> **Reply:** We reviewed the results and corrected areas where upper bounds were unnecessarily loose $(\text{rank}(C\_t - \nabla f(W\_t)) \leq n)$. The remaining $+n$ terms arise from $\Vert O_t \Vert\_\rm{F}^2 \leq n$, which cannot be avoided under the current proof strategy. We acknowledge this limitation and have added a note to the manuscript (see the paragraph below Theorem 3.1).
>
> ---
>
> **Requested Change 3** Either incorporate Newton–Schulz truncation error in the theory, or clearly scope the results to an idealized algorithm.
>
> **Reply:** Following your reasonable comment, we have further clarified the possibility of a gap between the ideal muon we are theoretically considering and the practical one (See the last part of Section 2.1).
>
> ---
>
> **Requested Change 4-1:** Rework the critical batch size derivation so it does not rely on discarding $Z$ (Section 4.1).
>
> **Reply:**
> Due to space constraints, please refer to the Reply to Weakness 2-1 to Reviewer **d7rw**.
>
> ---
>
> **Requested Change 4-2:** Align experiments with the theoretical stopping proxy (average gradient norm).
>
> **Reply:** We agree that the experiments should better match the theoretical stopping proxy (an upper bound on the average expected full-gradient norm). In the revision, we add a gradient-norm–based diagnostic:
>
> - We define a theory-aligned stopping rule using the Frobenius norm of the (mini-batch) gradient, with EMA smoothing:
>   $g_t=\Vert \nabla f\_{\mathcal{S}\_t}(W\_t)\Vert\_\rm{F}$, stop at the first step where $\tilde g_t \le \bar \epsilon_{\text{tot}}$.
> - We report **steps** $T_\bar \epsilon(b)$ and **SFO** $b\cdot T_\bar \epsilon(b)$ to reach a fixed gradient-norm threshold $\bar \epsilon_{\text{tot}}$.
> - We keep the original accuracy/loss targets for practitioner-facing comparisons, but explicitly position the gradient-norm diagnostic as the one used to validate the theory-aligned proxy.
>
> **Where in the paper.**
> Added to Section 5 (Numerical Experiments): a new paragraph “Theory-aligned stopping proxy: gradient-norm threshold”, and additional CBS results under this criterion in Appendix (see “Additional Diagnostics: Gradient-norm proxy and full-Muon toy setting”).
>
> ---
>
> **Requested Change 5:** Clarify novelty and positioning relative to existing Muon convergence analyses cited in Section 6.
>
> **Reply:** To address the reviewers' request, we added a simple table to Section 6 to clarify our novelty (see Table 3).
>
> ---
>
> **Requesetd Change 6:** Clarify the mismatch between the theoretical setting and the experimental optimizer.
>
> **Reply:** We clarify that the theory analyzes **full Muon**, whereas some practical-scale vision experiments use a **hybrid optimizer** (Muon on matrix-shaped blocks, AdamW or MomentumSGD on the remaining parameters), following common practice. To address this mismatch:
>
> - We explicitly state this as a limitation of direct theory <-> experiment correspondence for the vision workload.
> - We add a controlled **full-Muon MLP** experiment where Muon is applied to **all parameters**, and evaluate both:
>   (i) steps/SFO to reach a gradient-norm threshold, and
>   (ii) the quantitative proxy fit $X/T+Y/b+Z$ and the induced prediction of $b^\star$.
>
> **Where in the paper.**
> (1) Section 5 Experimental Setup: added an explicit sentence clarifying the hybrid setting and pointing to the controlled full-Muon diagnostic.
> (2) Appendix: added a dedicated diagnostics section describing the full-Muon MLP setting and results.

---

> > ### Comment · Reviewer_mpr6 · 2026-02-12
> > **Okay**
> >
> > Okay, I’m generally okay with the revised version. I believe meets TMLR standards.
> > Best,

---

> > > ### Author Response · Authors · 2026-02-16
> > >
> > > Thank you for your review and response. We are deeply grateful.

---

### Review · Reviewer_1aGJ · 2026-01-13

**Summary Of Contributions:**

This paper studies the convergence bounds and the critical batch size of the Muon optimizer. For the convergence bounds, the authors evaluate Muon across four settings, i.e., with and without Nesterov momentum, and with and without weight decay. One major finding of the paper is that the inclusion of weight decay leads to strictly tighter theoretical bounds and it also improves our understanding of the interplay between the weight decay coefficient and the learning rate. Furthermore, the authors also derive the critical batch size for Muon which minimizes the computational complexity.

Strengths:
- This work appears to be the first in the literature to study the convergence rates of Muon under these four settings, with high practical relevance.
- The study of critical batch sizes for Muon is also interesting, because we can expect it to be highly different from those for Adam for instance.

Weaknesses:
- Convergence analysis of this kind for Muon focuses on one weight matrix W, instead of all parameters of a neural network. It appears to me that it is quite non-trivial to make the analysis applicable to all weights, especially for the analysis of critical batch sizes. In particular, $L$ and $\sigma^2$ could be very different for different layers.
- For the language model workload experiment, the results will be more convincing if the same analysis is performed also on pre-training a larger model. This helps us understand how critical batch sizes would scale with the model size.

**Additional Comments:**

N/A

**Audience:**

Yes

**Audience Explanation:**

Muon is a deep learning optimizer that has aroused much attention and interest from the machine learning community recently. This work has deepened our theoretical understanding of Muon especially when Nesterov momentum and weight decay are used.

**Claims And Evidence:**

Yes

**Claims Explanation:**

This paper has provided rigorous convergence proofs of the claimed theoretical results. These theoretical results are empirically evaluated with numerical experiments of several vision and language modeling workloads.

**Requested Changes:**

The writing style of the paper requires much improvement. I list my requested changes below:
- The citation style of this paper is problematic and the use of \citep and \citet should be distinguished. See https://www.jmlr.org/format/formatting-errors.html Citations section for how to cite papers correctly. Please correct ALL instances accordingly.
- The instances of hyphens (- in LaTeX), en-dashes (--) and em-dashes (---) used in this paper are also problematic. E.g., page 2: compute-time (should be hyphen), Muon---with and without… (should be em-dash), etc. See also Misc section in the above link. Please also correct ALL instances accordingly.
- On page 3, NewtonSchulz5 instead of NewtonShultz5
- For Theorems 3.1 and 3.2, please write complete sentences and do not use short-hands like w/o and w/. Theorems are rigorous mathematical statements.
- Some inline formulas have weird spacing. For instance, the captions of Figure 1 and the convergence analysis paragraph on page 8. Can you check that?

- Caption of Figure 2: *the* fastest
- Below the caption of Figure 3: Nesterov *momentum*
- A lot of the references are incomplete. In particular, the arXiv papers all have their preprint number missing, and Petrov et al. (2025) has even an incomplete title. Please check the references carefully. Also, use {} in the bib file to properly capitalize letters such as {M}uon, {LLM}, etc.

---

> ### Author Response · Authors · 2026-01-26
>
> We deeply appreciate your thorough review. We have prepared the following rebuttal and kindly request your review.
>
> ---
>
> **Response to the comment on model scaling**
>
> We thank the reviewer for the insightful suggestion. We agree that understanding how the critical batch size ($b^\star$) evolves with model size is of significant practical interest.
>
> **1. Computational constraints for Critical Batch Size analysis**
> We would like to highlight that determining the critical batch size is significantly more computationally intensive than a standard training run. It requires measuring the SFO complexity curve by performing a grid search over multiple batch sizes, and critically, tuning the learning rate for each batch size. Scaling this analysis to significantly larger models (e.g., multi-billion parameters) was unfortunately infeasible given our academic computational budget.
>
> **2. Consistency in preliminary scale-up**
> However, as a feasible verification within our resources, we extended our Llama experiments by scaling from 160M to 300M parameters. We observed that the qualitative trends regarding the trade-off between SFO complexity and batch size remained consistent with the results reported in the main text within limited number of new experiments. This suggests that the optimization dynamics described in our analysis hold within this regime.
>
> **3. Theoretical mechanism for scaling**
> While we could not empirically simulate massive scale pre-training, our theoretical analysis provides the mechanism for how scale affects batch size. Our results establish that the critical batch size is proportional to the gradient variance: $b^\star \propto \sigma^2$ (Proposition 4.3). This implies that scaling behaviors observed in larger models are mediated through changes in their gradient noise properties.
>
> To clarify this, we have revised Section in the updated manuscript. We explicitly discuss how our variance-based theory relates to model scale and cite recent large-scale empirical findings (e.g., Liu et al., 2025), which report that Muon remains efficient at large batch sizes for LLMs—an observation that aligns with our theoretical framework.
>
> ---
>
> **Response to the comment on single-matrix analysis vs. full network**
>
> We appreciate the reviewer pointing out the heterogeneity of layer statistics. We agree that in a deep neural network, the constants $X$ and $Y$ (related to gradient variance $\sigma^2$) naturally vary in magnitude across different layers.
>
> **1. Correlation of gradient noise across layers**
> While the absolute magnitudes of these statistics differ, recent empirical research supports the view that they evolve coherently across the network. Specifically, **Gray et al. (NeurIPS 2024)** investigated the Gradient Noise Scale (GNS)—a metric directly related to the critical batch size ($B_{\text{crit}} \approx \text{GNS}$)—in Transformer models. They found that the GNS of individual layer types (Attention, MLP, and Normalization) exhibits extremely high correlation with the GNS of the entire model, despite differences in scale (see Figure 7 and Section 4.2 in their paper).
>
> **2. Validity of single-matrix proxy**
> This high correlation suggests that while our single-matrix analysis cannot predict the exact integer value of the critical batch size for a full network without aggregating total variance, it effectively captures the fundamental dependencies of the critical batch size on hyperparameters. Since the noise statistics across layers move in synchronization, the functional relationship we derived—how $b^\star$ scales with momentum $\beta$ or weight decay $\lambda$—is expected to hold qualitatively for the global dynamics.
>
> We have added a discussion in Section 7 (Conclusion) citing Gray et al. (2024). We acknowledge the layer-wise heterogeneity as a limitation, while noting that the strong correlation of gradient noise across layers suggests our single-matrix analysis can effectively capture the qualitative scaling behavior of the critical batch size with respect to hyperparameters.
>
> ---
>
> **Reply to Requested Changes:** We apologize for any imperfections in the initial draft.
> We have addressed all your requested changes and revised the manuscript accordingly. We are grateful for your thorough and meticulous review of our paper.
>
> **Reference**
>
> Gray et al. (2024): https://arxiv.org/pdf/2411.00999
>
> Liu et al. (2025): https://arxiv.org/pdf/2502.16982

---

> > ### Comment · Reviewer_1aGJ · 2026-02-12
> >
> > Thank you. I am generally okay with the revised version.
> >
> > I have some more suggestions:
> > - In Table 3, use \citet instead of \citep for the paper citation.
> > - The paper Lau et al. (2025) also appears to have convergence rate analysis and it could be more comprehensive to include it in related work and Table 3.
> > - For easy reference, it is also a good idea to include the convergence rate this paper developed in Table 3.
> >
> > *Reference*
> > Lau et al. (2025): https://arxiv.org/abs/2505.21799

---

> > > ### Author Response · Authors · 2026-02-12
> > >
> > > Thank you for your response. We have addressed the points you raised in Table 3.
> > >
> > > We deeply appreciate your thorough review and your dedicated efforts in improving the manuscript.

---

> > > > ### Comment · Reviewer_1aGJ · 2026-02-12
> > > >
> > > > I think in addition to Table 3, adding a sentence in the main text discussing Lau et al. (2025) will make it more coherent, similar to other mentioned papers.

---

> > > > > ### Author Response · Authors · 2026-02-12
> > > > >
> > > > > Thank you for your prompt reply. We have mentioned the literature you pointed out in Section 6.

---

> > > > > > ### Comment · Reviewer_1aGJ · 2026-02-12
> > > > > >
> > > > > > Thanks for the quick revision.

---

### Decision · Action_Editor_XYeY · 2026-03-01

**Recommendation:** Reject

**Audience:**

Yes

**Audience Explanation:**

Convergence of Muon is a relevant topic of ML.

**Claims And Evidence:**

No

**Claims Explanation:**

There is lack of consensus among the reviewers about the merits of the paper and multiple reviewers have pointed out a key theoretical gap in the submission - that this is not a convergence result as is being claimed. Further, its concerning that the ``extra'' error term is scaling with the dimensions of the optimization parameter. So, the result is significantly missing the bar for it to be a proof of convergence.

Further, the argument about critical batch-size is not convincing - as multiple reviewers have pointed out. The newly added Remark 4.1, only makes the reviewer's criticism more clear -- the relevant quantity $\epsilon$ is almost always negative as $Z$ is huge while $\epsilon_{tot}$ is intended to be small.

The experiments are interesting and quite detailed but this paper is not designed to make that the main message the reader gets.

I would encourage the authors to consider a resubmission where (A) either the theoretical result is actually a convergence result or (B) the writing is framed as paper on empirical studies on Muon while the current theory is something ancillary in the appendix giving some motivations for the experiments.

**Resubmission Of Major Revision:**

The authors may consider submitting a major revision at a later time.